# Fast and Slow Generating: An Empirical Study on Large and Small Language Models Collaborative Decoding

## Abstract

Large Language Models (LLMs) exhibit impressive capabilities across various applications but encounter substantial challenges such as high inference latency, considerable training costs, and the generation of hallucinations. Collaborative decoding between large and small language models (SLMs) presents a promising strategy to mitigate these issues through methods including speculative decoding, contrastive decoding, and emulator or proxy fine-tuning. However, the specifics of such collaborations, particularly from a unified perspective, remain largely unexplored. Inspired by dual-process cognitive theory, we propose a unified framework in this paper, termed Fast and Slow Generating (FS-GEN). Within this framework, LLMs (sometimes along with SLMs) are categorized as System 2 (slow and deliberate), while independent SLMs are designated as System 1 (fast and intuitive). We provide a comprehensive analysis of these collaborative methodologies, elucidating their common properties and shedding light on the differential knowledge capabilities of System 2 versus System 1 through the FS-GEN framework. Our findings indicate that only a small proportion of collaborative interactions (approximately less than 20% in most instances) are necessary across various methods. These interactions between System 1 and System 2 conform to a scaling law related to the parameter ratios, enabling predictable collaboration. Furthermore, we explore the specific conditions under which collaboration proves most effective, particularly from an uncertainty perspective, offering novel insights that may guide future optimization efforts. Our research underscores that the fundamental distinction between System 1 and System 2 lies in the uncertainty of next token predictions, where interventions by System 2 are crucial to support System 1. We provide code for reproduction: https://anonymous.4open.science/r/ICLR2025_Anonymous-127D

## 1 Introduction

In recent years, the landscape of artificial intelligence has been dramatically reshaped by the development of Large Language Models (LLMs) like GPT-4 (Achiam et al., 2023), Gemini (Team et al., 2023), Llama (Touvron et al., 2023a;b; AI@Meta, 2024), and Qwen (Bai et al., 2023). These models have emerged as versatile tools, enhancing software development (Qian et al., 2023; Yang et al., 2024), and accelerating scientific discovery (AI4Science & Quantum, 2023; Qi et al., 2023) in different domains (Boiko et al., 2023; Romera-Paredes et al., 2024). Despite their advancements, these models face significant challenges that affect their practical application. Key issues include managing inference time with constraints such as low-latency requirements (Miao et al., 2023; Zhou et al., 2024), and addressing concerns related to hallucinations in generated content (Huang et al., 2023). Furthermore, the training of these models entails considerable resource consumption (Hu et al., 2021; Hoffmann et al., 2022; Ding et al., 2023), particularly during fine-tuning and in achieving domain specialization (Bommasani et al., 2021).

A novel trend in mitigating these challenges involves the collaborative decoding of large and small models. This approach leverages the strengths of both model sizes to improve efficiency and effectiveness across various downstream applications. In these methods, the next token during generation depends on both LLMs and SLMs, whose logits are combined to form the final predictions. For in-

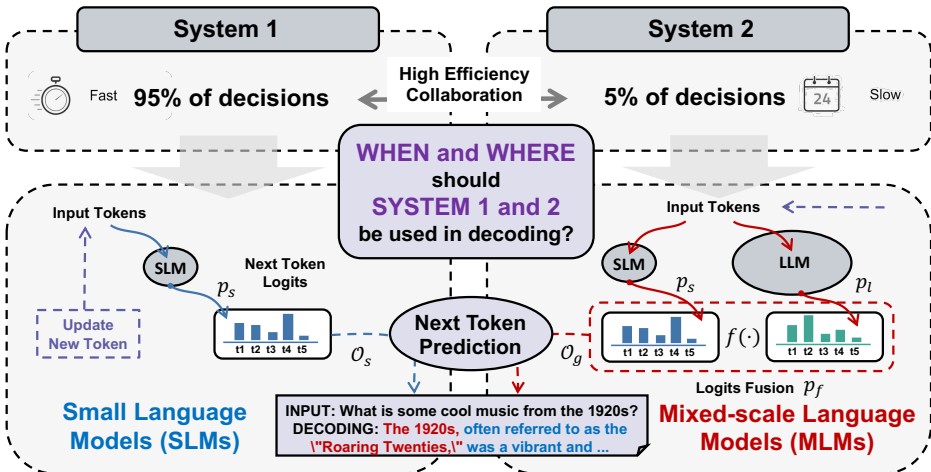

Figure 1: In the fast-and-slow generation framework, we conceptualize small language models as System 1 and a mixed-scale compound system of large and small language models as System 2. This compound system incorporates various collaborative decoding methods, such as speculative decoding, contrastive decoding, and proxy tuning. In this paper, we investigate the optimal conditions and contexts for applying System 1 and System 2 at the token-level decoding stage.

stance, speculative decoding (Leviathan et al., 2023; Kim et al., 2024; Sun et al., 2024; Xia et al., 2024) allows small models to swiftly generate initial drafts, which are then refined by large models, significantly cutting down on inference time. Similarly, in contrastive decoding (Li et al., 2022; O'Brien & Lewis, 2023; Sennrich et al., 2023), small models contribute amateur tokens as negative signals to enhance the decoding. Proxy tuning (Mitchell et al., 2023; Liu et al., 2024a) aims to reduce training costs by applying the delta logits from proxy small models to the base large models, rather than directly tuning the large models. This method significantly reduces training costs.

However, while the use of large and small models in tandem is increasingly common, the research community lacks a unified framework to analyze and understand the dynamics of their collaboration. This gap in knowledge motivates our current study, which seeks to formalize the collaboration between these models from the perspective of Systems 1 and System 2—a framework typically applied to modeling human cognitive processes (Kahneman, 2011). As shown in Figure 1, within the human brain, System 1 operates quickly and intuitively, automatically addressing everyday problems such as pattern recognition. Conversely, System 2 functions in a slower, more deliberate manner, tackling complex issues that require long-term planning. The integration of System 1 and System 2 in daily human activities achieves maximum efficiency and demonstrates robust intelligence. Notably, System 1 handles approximately 95% of everyday tasks, while System 2 is responsible for the remaining 5% (Daniel, 2017). The fast and slow thinking paradigm has been applied in various LLM applications (Hagendorff et al., 2023; Lin et al., 2024; Qi et al., 2024), which focus on improving the abilities of System 2 and the inference efficiency of LLMs. This paradigm provides a human-like mechanism for model optimization toward advanced artificial intelligence. Similarly, large and small models exhibit the natural characteristics of slow and fast processing. Previous works (Kim et al., 2024) on speculative decoding indicate that only a small percentage of generations require intervention by LLMs, while SLMs can independently complete most of the remainder.

However, many unknowns persist in these collaborations within this broader paradigm, including questions about when and where System 2 should be utilized. To the best of our knowledge, no previous work has attempted to unify the collaborative decoding methods of SLMs and LLMs from this perspectives. In this paper, we explore the fundamental questions of model collaboration under the fast and slow generation framework: *1) What is the essential difference between Systems 2 and 1 from the perspective of decoding? 2) Why does System 1 under-perform compared to System 2? 3) How can we improve effiency of the large and small compound system?* In this paper, we focus exclusively on the high-efficiency features of System 1 and System 2 based on language models, deferring an exploration of their in-depth reasoning capabilities to future work.

Table 1: Motivation, roles of SLMs and LLMs in different collaborative decoding methods.

| Method | Speculative Decoding | Contrastive Decoding | Proxy Tuning |
|---|---|---|---|
| Motivation | Inference efficiency | Factuality and Reasoning | Training efficiency |
| Role of SLMs | Drafter (generates candidate tokens) | Amateur (generates suboptimal tokens) | Domain expert (provides delta logits) |
| Role of LLMs | Verifier of draft tokens | Expert generator | Base logits provider |
| System 1 | SLMs | SLMs | SLMs |
| System 2 | LLMs | LLMs + SLMs | LLMs + SLMs (FT/Base) |
| Collaborations | Generation-verification | Logits fusion | Logits fusion |

To investigate these problems, we focus on analyzing the frequency and position characteristics of collaboration between large and small models, employing both qualitative and quantitative methods. Our research examines theoretical limits to collaboration frequency and employs scaling laws to predict how often different model combinations might collaborate based on the disparity in their parameters. Additionally, we explore the position tendencies of these collaborations, particularly in terms of where in the generation process they are most effective and how they correlate with the inherent uncertainty of small models. Our preliminary findings are summarized as follows:

- A comparison between the collaboration dynamics in large and small models and human dual-system processes reveals a shift from the expected 95:5 fast and slow ratio to an 80-20 distribution (§ 5.1.1). This finding also suggests a strong correlation between the frequency of interventions and the size differences in parameters between models (§ 5.1.2), in accordance with scaling laws.

- Collaborative interactions among models are most critical at the beginning of the generation process, underscoring the adage that "Well begun is half done." (§ 5.2.1) The uncertainty levels in next token prediction of smaller models play a pivotal role, serving as a reliable indicator of collaboration necessity (§ 5.2.2).

## 2 RELATION WORKS

**Collaborative Decoding Methods.** Collaborative decoding has emerged as an effective strategy to enhance large language models (LLMs) by leveraging both high-performing LLMs and cost-efficient small language models (SLMs). Speculative decoding accelerates the generation process by having SLMs draft candidate spans, which are then verified by LLMs in a single forward pass, with verified tokens fed back for subsequent generations (Leviathan et al., 2023; Kim et al., 2024; Sun et al., 2024; Xia et al., 2024). This method focuses on drafting robust candidates (Xia et al., 2023; Liu et al., 2024b) and efficient verification strategies, such as lossless and approximate approaches for greedy decoding and nucleus sampling (Stern et al., 2018; Xia et al., 2023; Fu et al., 2023b). Contrastive decoding, on the other hand, aims to improve factuality (Sennrich et al., 2023) and reasoning abilities (O'Brien & Lewis, 2023) by fusing logits from expert and amateur models (Li et al., 2022), under the assumption that amateur models' outputs are less reliable. This approach enhances scrutiny on the tokens generated, reducing hallucinations and improving output quality (Jin et al., 2024). Both methods primarily explore how to fuse logits without fully addressing the extent and specific needs for collaboration. Additional techniques like emulator or proxy tuning involve multiple models to improve training efficiency (Mitchell et al., 2023; Liu et al., 2024a) but similarly lack deeper empirical analysis on collaboration dynamics.

**Scaling Laws and Emergent Abilities.** Scaling laws have been instrumental in the development of LLMs, indicating that emergent abilities often manifest in larger models while smaller models struggle with the same tasks (Kaplan et al., 2020; Hoffmann et al., 2022; Wei et al., 2022). Recent studies suggest that these abilities are influenced by the continuity of evaluation metrics: continuous metrics show minimal divergence between model sizes, whereas discrete metrics reveal significant variations (Schaeffer et al., 2024; Hu et al., 2024). Despite these insights, there remains a lack of empirical analysis into the essential differences between LLMs and SLMs from a decoding perspec-

tive. This paper addresses this gap by exploring these differences through collaborative decoding, providing new insights that significantly enhance the efficiency and quality of existing techniques.

## 3 PRELIMINARY

### 3.1 TEXT GENERATION

Text generation, also referred to as decoding (Welleck et al., 2024), is the process whereby a language model $\mathcal{M}$ selects the next token in a sequence based on the context. Mathematically, this involves estimating the conditional probability of a token given its preceding sequence. For a given source context with $N$ tokens and a target response with $T$ tokens, noted as $X = (x_1, x_2, \cdots, x_N)$ and $Y = (y_1, y_2, \cdots, y_T)$, the probability of the response sequence is expressed as:

$$P(Y|X) = \prod_{i=1}^{T} p(y_i|X, y_1, y_2, \cdots, y_{i-1})$$

During the decoding phase, at each step $i$, the model $\mathcal{M}$ computes the probability distribution $p(y_i|X, y_1, y_2, \cdots, y_{i-1})$ over the entire vocabulary, based on the input sequence provided so far. The selection of the next token $y_i$ is then based on this distribution, typically employing methods such as greedy search (Parsing, 2009), beam search (Vijayakumar et al., 2016; Freitag & Al-Onaizan, 2017), or various sampling strategies (Fan et al., 2018; Holtzman et al., 2020).

### 3.2 DEFINITION OF FS-GEN METHODS

Beginning with the fundamental definition of text decoding, we extend this concept to encompass collaborative decoding. In order to provide a unified definition of the collaborations employed in various FS-GEN methods, we consider setups involving large language models, denoted as $\mathcal{M}_l$, and small models, denoted as $\mathcal{M}_s$. The respective input prompts for $\mathcal{M}_l$ and $\mathcal{M}_s$ are given as $X_l$ and $X_s$, which are identical in most settings. As illustrated in Figure 1, the logits for the next token generated by LLMs and SLMs are denoted as $p_l$ and $p_s$, respectively, with the reference golden logits noted as $p_f$. We categorize the outputs into three types: golden or collaborative outputs $\mathcal{O}_g$, which can also be denoted as $\mathcal{O}_g$, representing fused outputs., outputs $\mathcal{O}_l$ from $\mathcal{M}_l$, and outputs $\mathcal{O}_s$ from $\mathcal{M}_s$. Note that the term golden logits $p_f$ and outputs $\mathcal{O}_g$ refers to the results of collaboration, which may not always equate to the golden answers in some instances. These variables are assigned differently across different methods, varying according to the fusing strategy and underlying motivation:

**Speculative Decoding.** This approach leverages predictions from SLMs that are subsequently validated or refined by LLMs. In this setting, the outputs from LLMs are considered golden outputs, denoted as $p_f = p_l$ and $\mathcal{O}_g = \mathcal{O}_l$, where $\mathcal{M}_l$ signifies the model chosen to select tokens, reflecting the trust in the outputs $\mathcal{O}_s$ from $\mathcal{M}_s$. Speculative decoding represents the most essential case.

**Contrastive Decoding.** This method employs differential strategies to reconcile the disparities between the predictions from large and small models, aiming to enhance accuracy. In the implementation (O'Brien & Lewis, 2023), the reference logits are calculated as $p_f = p_l + \beta(p_l - p_s)$, and $\mathcal{O}_g$ outperforms both $\mathcal{O}_l$ and $\mathcal{O}_s$, indicating that $\mathcal{O}_g$ leverages the strengths of both models.

**Proxy Tuning.** This strategy involves tuning SLMs to emulate the behavior of LLMs, thus minimizing training resource usage without significantly compromising output quality. In this configuration, SLMs are differentiated into base and chat versions, with their logits denoted respectively as $p_s$ for the base version and $p'_s$ for the chat version. The fused logits are represented by $p_f = p_l + (p'_s - p_s)$, suggesting that $\mathcal{O}_g$, incorporating insights from both $\mathcal{O}_l$ and the delta change between the tuned and untuned states of SLMs, approaches the performance of the fully tuned LLMs.

As shown in Table 1, in speculative decoding, SLMs serve as the base models for fast generation, while LLMs manage the more deliberate and slow aspects, providing interventions to the fast process. In contrast, there is difference in contrastive decoding and proxy tuning, with SLMs introducing interventions into the LLM generation process. In addition to the three typical methods, further exploration of other FS-GEN methods is displayed in Appendix B.

### 3.3 DEFINITION OF COLLABORATION FREQUENCY

As shown in Figure 1 and as discussed previously, not all steps in the fast system's generation require collaborations or interventions from the slow system. We denote the number of collaborations in FS-GEN methods as $N_l$ and total length of generated content as $N_{total}$. We defined collaboration frequency as $CoF = \frac{N_l}{N_{total}}$, which is typically set to 1 in contrastive decoding and proxy tuning. We can establish a lower bound of collaboration frequency, denoted as $CoF_{lower}$, which has been preliminarily proven in prior research (Kim et al., 2024) and is further discussed in § 5.1.1. The term "lower bound" denotes the minimal collaboration rate required for the text produced by SLMs (System 1) to exactly match the output generated solely by SLMs and LLMs (System 2). We provide a running example for computing $CoF_{lower}$ in Appendix C for better understanding on our settings.

## 4 EMPIRICAL STUDY

### 4.1 EXPERIMENT SETUP

**Models.** Our selection of language models primarily includes the Qwen (Bai et al., 2023) series and Pythia series (Biderman et al., 2023), and OpenELM series (Mehta et al., 2024) which feature the broadest range of model parameters. The Qwen series spans from 0.5B to 72B, while the Pythia series covers from 70M to 12B. For the FS-GEN methods, model combinations are strategically grouped based on size to facilitate collaborative decoding, such as pairing 0.5B with 14B, and 7B with 72B. To validate and observe the original behavior of models, we utilize the base models in addition to implementing proxy tuning, which involves combining base and chat models. For the Pythia series, we fine-tuned the base models using the `hkust-nlp/deita-10k-v0` dataset (Liu et al., 2023) to develop a chat version.

**Datasets.** To explore the common features of collaborative decoding across various tasks, our experiments utilize datasets tailored to distinct domains: MMLU-STEM (Hendrycks et al., 2021) for general knowledge, GSM8k (Cobbe et al., 2021) for math, and MBPP (Austin et al., 2021) for coding. Analysis is primarily conducted on the test sets to mitigate the risk of training data leakage; however, we also perform statistical analyses on the training data for threshold setting.

Our experiments are conducted exclusively using greedy decoding, which effectively captures the fundamental uncertainty associated with tokens in both large and small models (Wang & Zhou, 2024). Drawing on previous studies involving speculative and contrastive decoding, greedy search could potentially be adapted to sampling decoding through temperature rescaling. (Leviathan et al., 2023; Chen et al., 2023; 2024). Additional implementation details are provided in Appendix D.

### 4.2 EXPERIMENT DESIGN

Our empirical study investigates two key aspects of collaboration in FS-GEN: frequency and position. Collaboration frequency evaluates the minimum necessary interactions between large and small language models, enabling predictions of interaction frequency across different model scales. Collaboration position identifies specific interaction points, predicting when collaboration is needed. For collaboration frequency, we first analyze lower bound of frequency across various model scales and tasks to identify trends (§ 5.1.1). Next, we examine a scaling law in FS-GEN that links models scale ratios with optimal interaction frequency for prediction (§ 5.1.2). For collaboration position, we initially study mismatch points relative to generated content length (§ 5.2.1), followed by an analysis based on token uncertainty in SLMs (§ 5.2.2).

## 5 EXPERIMENTAL RESULTS

### 5.1 EXPLORING COLLABORATION FREQUENCY

We explore collaboration frequency through analyses of the lower bound and scaling laws. In the former analysis, we compute the lower bound of collaboration frequency for models of different scales across tasks. In the latter analysis, we examine the relationship between the lower bound frequency and the models' parameters ratios, which aids in building predictable collaborations.

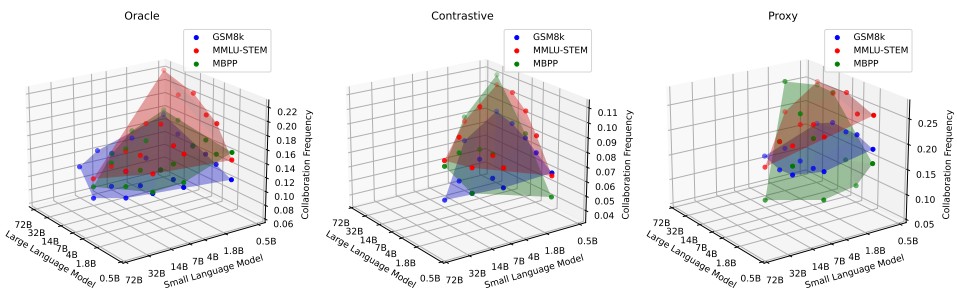

Figure 2: Collaboration frequency between models across various tasks (**Qwen Series**).

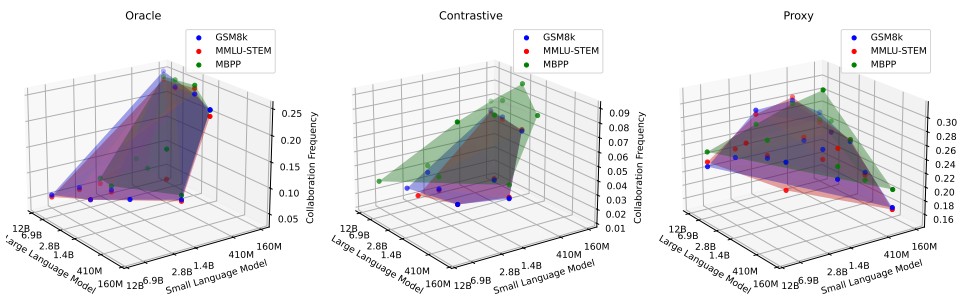

Figure 3: Collaboration frequency between models across various tasks (**Pythia Series**).

### 5.1.1 WHAT IS THE LOWER BOUND OF THE NUMBER OF COLLABORATIONS IN FS-GEN METHODS?

> ***Finding 1: 20% Collaborations (2:8 Law)***
>
> The lower bound of collaboration frequency between models of different scales averages less than 20% on various tasks and methods, reflecting the 2:8 law (the Pareto Principle).

As illustrated in Figure 2 and 3, the collaboration frequency varies with the difference between the task and model combination. Notably, for the specific SLMs, the frequency of collaboration increases as the size of LLMs increases, while models of similar sizes exhibit lower frequencies. When the parameters of the models are more closely aligned, the $CoF_{\text{lower}}$ is reduced, generally not exceeding 20% across various tasks. This observation suggests that the parameter gap between models does not conform to traditional scaling laws, where task performance is typically emergent (Wei et al., 2022). It also indicates that scaling laws may be influenced by the continuity of metrics (Schaeffer et al., 2024), which are affected by token uncertainty. The primary distinction between large and small models lies in the selection of essential tokens. Small models can be significantly enhanced by a few key tokens from larger models, particularly those that are difficult for them to select. This phenomenon is evident not only in oracle decoding, a form of speculative reasoning, but also in other collaborative decoding methods such as contrastive and proxy decoding.

### 5.1.2 WHAT IS THE RELATIONSHIP BETWEEN MODELS' RATIOS AND LOWER BOUNDS?

> ***Finding 2: Parameters Scale Ratio Law***
>
> The frequency of collaboration between a large and a small model during token generation is inversely proportional to the ratio of their parameters raised to a specific power.

Inspired by previous works on the "Scaling Law" (Kaplan et al., 2020; Ruan et al., 2024; Besiroglu et al., 2024; Hu et al., 2024), we aim to develop a scaling law relating the models' parameter ratios to the lower bound of collaboration frequency ($CoF_{\text{lower}}$). Let number of parameters of $\mathcal{M}_l$ and $\mathcal{M}_s$ are denoted as $\mathcal{N}_l$ and $\mathcal{N}_s$ respectively. Intuitively, the $CoF_{\text{lower}}$ is inversely proportional to the

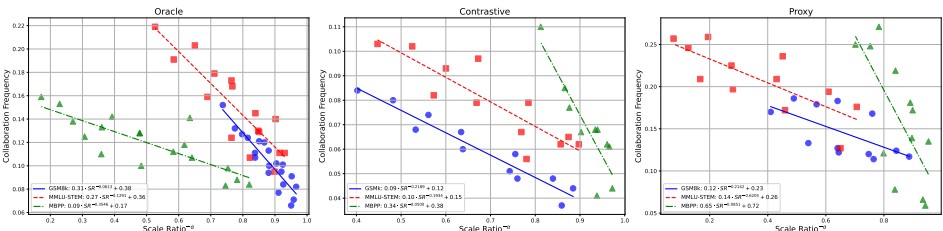

Figure 4: The Fitting Line of Parameters Scale Ratio Law for Qwen Series.

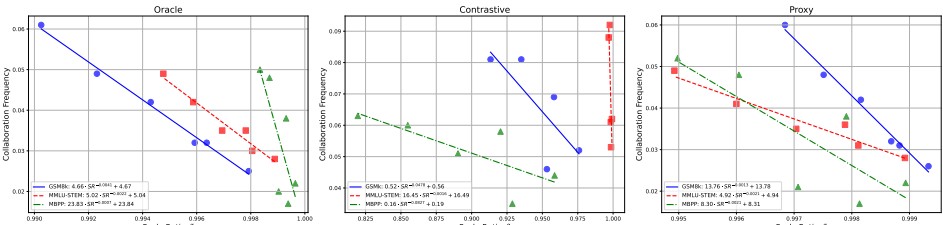

Figure 5: The Fitting Line of Parameters Scale Ratio Law for OpenELM Series.

scale ratio $R = \frac{\mathcal{N}_l}{\mathcal{N}_s}$. The relationship between the variables can be expressed by the equation:

$$CoF_{\text{lower}} = \gamma \cdot R^{-\alpha} + \beta$$

where $\gamma$ is a negative constant that determines the sensitivity of the $CoF_{\text{lower}}$ to the scale ratio, $\alpha$ is the scaling exponent, $\beta$ is a fitting parameter.

As the scale ratio $R$ increases, the collaboration frequency also increases. This is because the distance between the output distributions of the larger model and the smaller model increases (as "Scaling Law" works), necessitating a higher proportion of collaboration $CoF_{\text{lower}}$ from the smaller model. Figure 4 and 5 illustrate the fitting curves for different $R$ across various tasks and FS-GEN methods. The results indicate that the lower bound of collaboration frequency is related to the scale ratio between LLMs and SLMs, as well as the specific tasks and methods involved.

We find that the fitting line is significantly influenced by the models' performance, with better-fitting accuracy observed in models with higher performance. Additionally, we provide the fitting line for the Pythia series in the appendix, which is challenging to fit due to its weaker performance.

## 5.2 EXPLORING COLLABORATION POSITION

Building on previous analysis, we examine the mismatch tokens and key collaboration positions. This includes analyzing the mismatch positions within sequences in § 5.2.1 and the relationship between mismatch positions and token uncertainty in § 5.2.2.

### 5.2.1 WHICH PART OF THE GENERATION MOST REQUIRES COLLABORATION OR INTERVENTION?

> **Finding 3: "Well begun is half done"**
>
> As the adage suggests, collaborations are most necessary at the beginning of the generation process, especially in reasoning tasks such as mathematics and coding.

As discussed, interventions from LLMs are necessary only at mismatch positions in generations between SLMs and LLMs. As depicted in Figure 6, 7, 8, and 13, we quantify these positions using a mismatch rate for all positions, analyzed with a 10% split where deeper colors indicate higher percentages and lighter colors represent lower percentages. Our initial analysis of mismatch positions across the entire generated content corroborates the importance of the generation's start. Prior

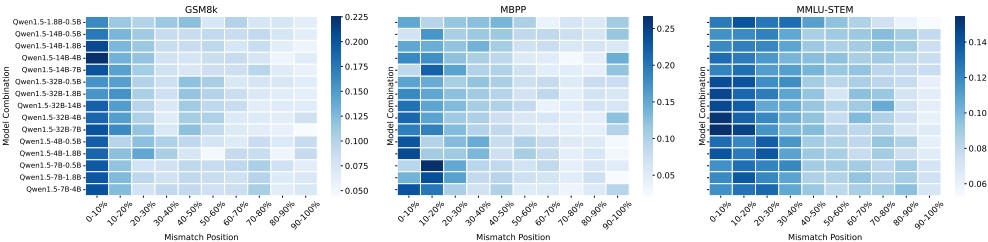

Figure 6: Percentage of mismatch positions relative to length for model combinations across various datasets (**Qwen Series, Oracle Decoding**).

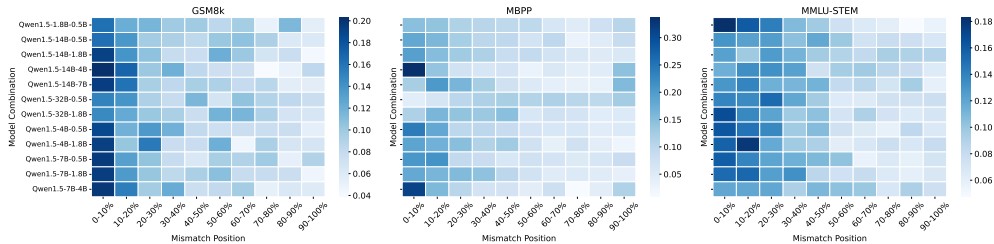

Figure 7: Percentage of mismatch positions relative to length for model combinations across various datasets (**Qwen Series, Contrastive Decoding**).

research on reasoning tasks demonstrates that the pass rate for smaller models can be significantly improved by substituting the initial steps with those generated by larger models (Jain et al., 2024; Wang & Zhou, 2024). This phenomenon is similarly observed during the collaborative decoding between large and small models. The findings reinforce the adage that a good start is crucial, largely due to the presence of more reliably conditioned tokens from larger language models. This effect is particularly pronounced in GSM8k and MBPP tasks, illustrating that steps are more critical in reasoning tasks than in common knowledge tasks. Additionally, this finding could also support privacy protection in cloud-device collaborations, as highlighted in recent work (Zhang et al., 2024).

### 5.2.2 WHICH POSITION IN FAST GENERATION MOST REQUIRES COLLABORATION/INTERVENTION?

> **Finding 4: Lost in Uncertain Tokens**
>
> Tokens exhibiting high uncertainty within the fast system are likely to require collaboration or intervention from the slow system.

Beyond a macro analysis of whole-length position discrepancies, we also delve into the mismatch token logits and internal uncertainty within the vocabulary. Next token prediction (Radford et al., 2018; 2019; Brown et al., 2020; Achiam et al., 2023), a cornerstone of the "Scaling Law", further highlights differences between large and small models. As depicted in Figure 9, this illustration shows that only a few tokens during the generation of a problem in the GSM8k task by small models require guidance from larger models, which are identified as red tokens. Compared to large models, small models often struggle with uncertain tokens, which leads to incorrect responses. These tokens exhibit a similar distribution, characterized by high uncertainty at each step, with particularly low confidence in the top token, which helps identify crucial collaboration points. These findings have become fundamental principles for routing draft models in previous works (Kim et al., 2024; Du et al., 2024) on speculative decoding. We also observe their occurrence in various FS-GEN methods (i.e., contrastive decoding and proxy tuning), as detailed in Appendix E.2.

To further investigate the uncertainty of token logits, we conduct a visual analysis of the top-k token logits at each generative step, noting whether they match. The resulting correlation plots are displayed in Figures 10 and 16. There is a high correlation between the mismatch positions

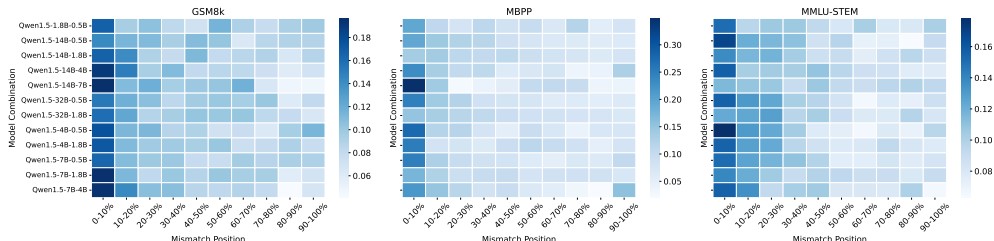

Figure 8: Percentage of mismatch positions relative to length for model combinations across various datasets (**Qwen Series, Proxy Tuning**).

**Question:** Colby works for a manufacturing company in the packaging division. He gets paid $0.20 for every package he completes. If he completes 10 less than 50 packages per hour, how much money, in dollars, does he earn in a typical eight-hour workday? Let's think step by step
**SLM Results(Wrong):** Colby works 50 - 10 = 40 packages per hour.\n He completes 10 less than 50 packages per hour, so he completes 40 - 10 = 30 packages per hour.\n He earns $0.20 for every package he completes, so he earns 30 * $0.20 = $6.\n In a typical eight-hour workday, he earns $6 * 8 = $48.\n The answer is 48
**Collaborative Decoding Results(Correct):**

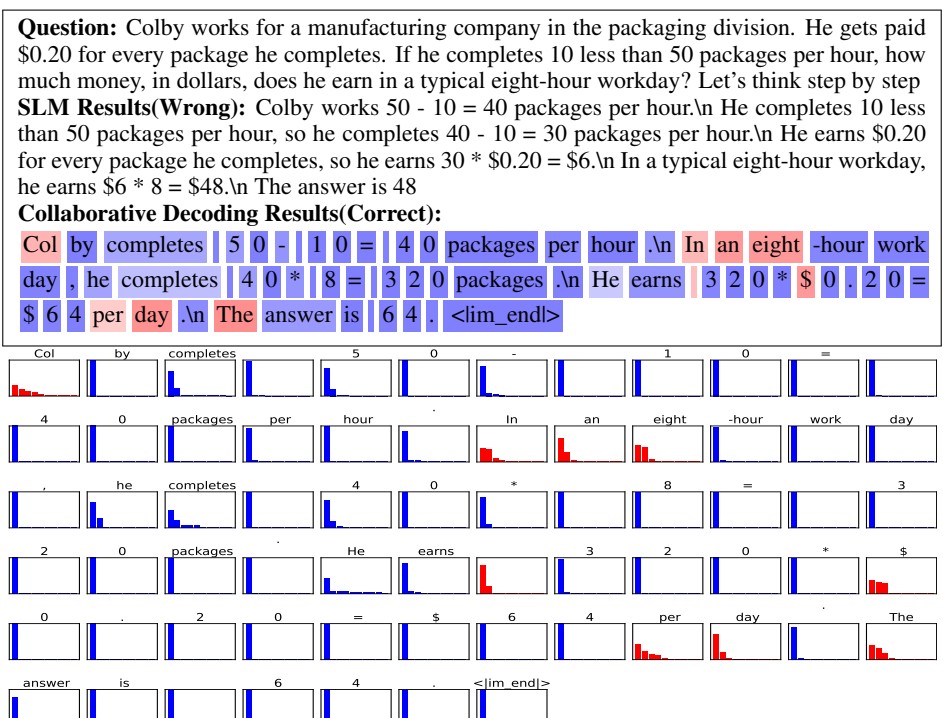

Figure 9: Visualization of token uncertainty in SLMs and LLMs. Red signifies mismatch tokens, while Blue indicates match tokens; darker shades represent higher confidence, where only with interventions from LLMs on red tokens can SLMs provide the correct answers.

and token uncertainty, particularly when more context tokens are considered. Additional cases and implementation details are provided in Appendix E.2.

# 6 DISCUSSION

**Difference in Various Tasks and Models.** While subtle differences in results may arise from variations in tasks and model capabilities, these differences do not impact our primary findings. The results are primarily influenced by two factors: data size and model performance: 1) Data Size. Due to computational constraints, we sampled approximately 500 data points for each task. This limited sampling may introduce fluctuations in the observed curves. 2) Model Performance. Parameter ratio scaling laws are strongly influenced by model performance. While Qwen series models exhibit consistent performance, Pythia models underperform due to insufficient pretraining, which affects the collaboration dynamics between large and small models.

**Cost-aware Collaboration Optimization.** In our experiments, we illustrate the lower bound of collaboration frequency; however, achieving this in practice is not fully feasible. We can optimize

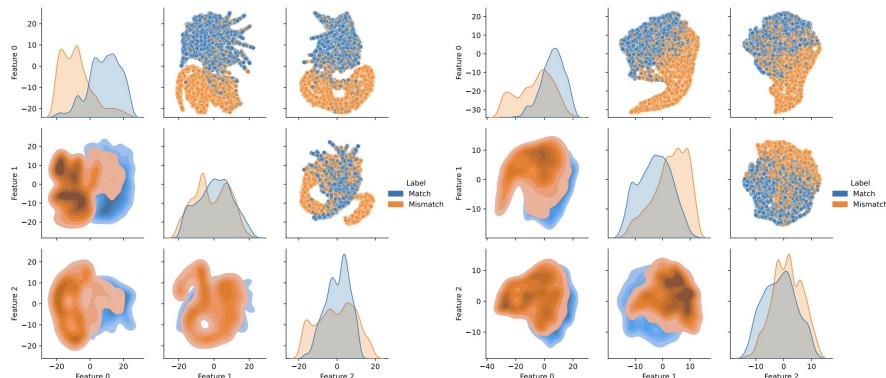

Figure 10: KDE and scatter visualization of logits distribution for mismatch/uncertain tokens from SLMs (**Qwen Series**). We provide the corresponding quantitative results in the Appendix F.1.

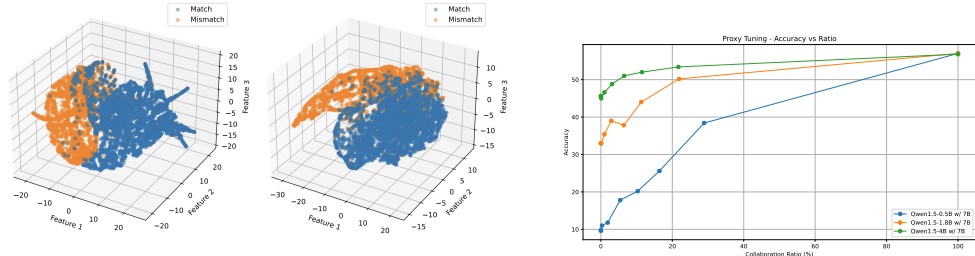

Figure 11: Left: 3D visualization of feature distribution for mismatch/uncertain tokens logits (1 & 5 tokens). Right: Routing with Top-1 Token Logits of SLM for Proxy Tuning (**Qwen Series**).

the collaboration frequency by examining token uncertainty, employing heuristic rules (Kim et al., 2024) that prove challenging to scale automatically across various scenarios. As shown in the right of Figure 11, we perform token-level routing based on threshold of the top-1 token logits from SLMs for proxy tuning, demonstrating promising efficiency-quality trade-offs for practical optimization. We use logits as features for routing, while other related metrics, such as entropy and perplexity-based logits, can be also explored. More analyses are provided in the Appendix F.1 and F.2. In conclusion, few studies consider these optimization of inference costs. Future research could explore strategies to balance inference costs and performance within a defined budget (Shen et al., 2024).

**More Model Series and Architecture.** In this paper, we exclusively conduct experiments on Qwen-series and Pythia-series models, which utilize transformer-based architecture. While further analysis could extend to different model series such as Llama and Yi (AI et al., 2024) models, the parameter scales of these models are not as extensive as those of the Qwen models. Additionally, recent state space models (SSM) (Gu et al., 2022) like Mamba (Gu & Dao, 2024; Dao & Gu, 2024) and hybrid architectures combining SSM and attention mechanisms, such as Jamba (Lieber et al., 2024), are garnering researcher interest. Speculative decoding, when applied to multimodal models (Gagrani et al., 2024), allows for the expansion of FS-GEN methods across various multimodal applications (Zhu et al., 2023; Team, 2024) Our findings can be further verified on these models.

# 7 CONCLUSION

In conclusion, our study provides a comprehensive analysis of collaborative decoding techniques under the fast and slow generation paradigm. By examining speculative decoding, contrastive decoding, and emulator fine-tuning, we identify key performance factors and conduct empirical analyses on collaboration frequency and positions. Our results demonstrate that the essential difference between large and small models lies in the uncertainty confidence of next token predictions during decoding. We hope this work will inspire further research and innovation in model collaboration.

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

# A    RELATED WORKS

## A.1    COLLABORATIVE DECODING

**Speculative Decoding.** Leveraging the high performance of LLMs and the low inference cost of SLMs, speculative decoding (Leviathan et al., 2023; Kim et al., 2024; Sun et al., 2024) has become an effective method to accelerate the generation process in LLMs. This approach involves two main steps: drafting and verification (Xia et al., 2024). In the drafting step, SLMs generate a candidate span comprising several tokens. Subsequently, in the verification step, LLMs review all draft tokens in a single forward pass. The verified tokens are then fed back to the SLMs as input for subsequent generations. Current efforts in this area can be categorized into two primary streams: drafting and verification. In drafting, researchers focus on developing robust drafters that aim to generate as many tokens as possible that are likely to be verified successfully (Xia et al., 2023; Liu et al., 2024b). For verification, methods such as lossless and approximate approaches (Stern et al., 2018; Xia et al., 2023; Fu et al., 2023b) have been proposed for both greedy decoding and nucleus sampling. Nearly all the aforementioned studies employ draft tokens of a fixed length, adjusted as hyper-parameters. However, recent efforts have begun to determine the candidate length dynamically (Mamou et al., 2024; Shen et al., 2024), with more in-depth empirical studies also currently under exploration.

**Contrastive Decoding.** Compared to the efficiency gains in speculative decoding, contrastive decoding (Li et al., 2022) focuses on improving factuality (Sennrich et al., 2023) and reasoning abilities (O'Brien & Lewis, 2023) through collaborative decoding. In this approach, the next token logits from expert and amateur models are fused comparatively. The underlying assumption is that tokens generated by the amateur model are less reliable and should be less frequently utilized in expert models, implying that greater scrutiny is required for tokens in expert models. Recent studies also explore contrasting models with and without context, designated as expert and amateur (Jin et al., 2024), which helps reduce hallucinations. Similar to speculative decoding, these efforts primarily examine how to fuse the logits, without addressing the extent and specific needs for collaboration.

**Emulator or Proxy Tuning and others.** Unlike enhancements in inference time that involve two models, emulator tuning (Mitchell et al., 2023) and proxy tuning (Liu et al., 2024a) employ three models to improve training efficiency. In these methods, researchers fine-tune only SLMs which generate logits from both base and chat model versions. By using the delta logits, the effectiveness of large chat models can be approximated through logits interpolation with large base models. Additionally, Shen et al. (2024) propose learning to collaboratively decode with multiple LLMs, including both general and specific domain models, to achieve domain generalization. Zhang et al. (2024) suggest a collaborative generation framework leveraging large and small models under cloud-device scenarios that focuses on secure context-aware instruction following.

**Scaling Law and Emergent Ability.** The experience of a "Scaling Law" has been successfully applied across various scenarios, including model size (Kaplan et al., 2020; Hoffmann et al., 2022), data size (Sorscher et al., 2023), and reward modeling (Gao et al., 2022; Rafailov et al., 2024), serving as a pivotal driver of LLMs development. Research indicates that emergent abilities (Wei et al., 2022) manifest in larger models as dictated by the scaling law, where certain capabilities appear exclusively in large models, with smaller models failing to solve the same problems effectively. In contrast, recent studies suggest that emergent abilities are significantly influenced by the continuity of indicators (Schaeffer et al., 2024; Hu et al., 2024). When continuous metrics such as token-level match are employed, minimal divergence is observed between large and small models. However, employing discrete metrics like accuracy in mathematical or reasoning tasks and pass rates on coding assignments reveals pronounced variations across models of different scales. This paper delves into the differences in abilities between large and small models from the decoding perspective, which can be quantified using continuous metrics. Through this lens, we explore the key distinctions driven by token uncertainty.

Totally, deeper empirical analysis is still lacking, and the quest for further efficiency optimization continues. This paper delves into the essential differences between LLMs and SLMs from the perspective of collaborative decoding. Our exploration provides new insights into various methods and significantly enhances the efficiency and quality of the aforementioned techniques.

# B  DEFINITION OF MORE FS-GEN METHODS

In accordance with specific definitions for various FS-GEN methods in § 3.2, we introduce additional variants as follows:

Mitchell et al. (2023) propose emulator tuning, which, like proxy tuning, is motivated similarly but differs in implementation. In this configuration, the SLMs are also divided into a base version and a chat version, with their logits represented as $p_s$ for the base version and $p'_s$ for the chat version. The fused logits are represented by $p_f = p_l \cdot \left( \frac{p'_s}{p_s} \right)$, indicating that $O_g$ combines the insights of the incremental changes between the tuned and untuned states of the SLMs, approaching the performance of a fully tuned LLMs.

Zhang et al. (2024) propose CoGenesis for secure, context-aware instruction following. In the logits-level variant of CoGenesis, the inputs for LLMs and SLMs differ, denoted as $X_l \neq X_s$. This approach utilizes predictions from both SLMs and LLMs, where SLMs generate personalized content and LLMs produce outlines. In this configuration, the reference logits are calculated as $p_f = f(p_l - p_s)$, where $f$ represents a trained model that combines features from both, and $\mathcal{O}_g$ outperforms both $\mathcal{O}_l$ and $\mathcal{O}_s$. This indicates that $\mathcal{O}_g$ effectively leverages the strengths of both models while preserving the privacy of user data.

# C  RUNNING EXAMPLE FOR COLLABORATIVE DECODING

To enhance understanding of our empirical study settings for collaborative decoding, we provide a running example with step-by-step verification. This example demonstrates the process of judging matches or mismatches in the generation of SLMs using generated context from mix-scaled models, as observed in the results of collaborative decoding.

Our primary objective is to analyze the frequency of collaboration in various decoding settings. In our research, we explore collaborative decoding (CoDec) at all steps ($CoF = 1$), for the given example in Table 2. For a lower collaboration frequency ($CoF_\text{lower}$), we input the outputs of CoDec into smaller models token by token to assess the consistency of top tokens. (CoDec represents speculative decoding ,contrastive decoding or proxy tuning). Assuming there are three mismatched tokens (e.g., "former", "78"), the calculated $CoF_\text{lower} = \frac{2}{18}$. However, unnecessary collaborations may occur even when matches are identified, leading to an variable where $CoF_{lower} \leq CoF \leq 1$. This motivates our investigation into the lower bounds of collaboration frequency, aiming to achieve similar outputs as full collaborative decoding with minimal collaborative steps. Our findings demonstrate this is a universal phenomenon across different collaborative decoding methods.

Speculative decoding currently selects a fixed number of tokens (K-tokens) for generation-verification, which does not effectively reach. In contrast, methods such as contrastive decoding and proxy tuning entail collaborations at each step ($CoF = 1$), which may not always be necessary.

Table 2: Running example for the computation of lower collaboration frequency.

**Input:**
User: "Who is Donald Trump?"
Assistant: "Donald Trump is the former President of the United States, who is 78 years old now."

**Computation:**
- First token verification, match:
    - CoDec: [Assistant]: "Donald"
    - Small: [Assistant]: "Donald"
- Second token verification, match:
    - CoDec: [Assistant]: "Donald Trump"
    - Small: [Assistant]: "Donald Trump"
- ...
- 5th token verification, mismatch:
    - CoDec: [Assistant]: "Donald Trump is the former President"

- Small: [Assistant]: "Donald Trump is President"
- ... (match)
- 14th token verification, mismatch:
  - CoDec: [Assistant]: "Donald Trump is the former President of the United States, who is 78"
  - Small: [Assistant]: "Donald Trump is the former President of the United States, who is 80"
- ... (match)

## D  IMPLEMENTATION DETAILS

In speculative decoding, the number of tokens generated by draft small models is set as a hyperparameter, denoted as $K$. In our empirical study, we define $K = 1$ as representing oracle decoding rather than speculative decoding. Although the experimental results are equivalent, implementing and analyzing $K = 1$ is simpler.

For each task across all methods, we adapt in-context few-shot to enhance generation performance under base models, using 5-shots for MMLU, 10-shots for MBPP, and 8-shots for GSM8k. The few-shot examples are unified into a chain-of-thought format (Fu et al., 2023a).

Note that we regard the outputs from collaborative decoding as references; therefore, we do not compute accuracy for each task. By default, results from collaborative decoding are generally better than those from either smaller or larger models alone. Our primary focus is on exploring the differences between larger and smaller models, rather than comparing them to the ground truth.

Additionally, our intent was to compare the top-1 token selected by the SLM and the collaborative models under greedy decoding. While the final token is obtained using $\arg\max$ logits, this approach is effectively equivalent to using probabilities after applying softmax. Therefore, we use logits throughout the main content, although probabilities could also be used for better understanding. For contrastive decoding, we primarily follow the implementation in (O'Brien & Lewis, 2023) rather than previous works (Li et al., 2022). This approach directly utilizes unnormalized scores (logits) assigned by the amateur and expert models. In current work, we use greedy decoding to compute the matching rate of tokens between small and large language models. This choice aligns with our initial motivation of achieving collaborative decoding with minimal intervention in small models, treating the collaborative decoding results as golden tokens. For scenarios where exact matching is less critical and the focus shifts to performance-speed optimization, other sampling techniques can be explored. These techniques might yield better performance with reduced collaboration frequency, leading to more efficient collaborations. However, quantifying results becomes more challenging due to the increased uncertainty introduced by sampling. We believe this is an exciting direction for future research, as it opens up possibilities for balancing efficiency and performance through alternative decoding strategies.

## E  MORE EMPIRICAL RESULTS

### E.1  RESULTS ON COLLABORATION FREQUENCY

To validate the robustness of our findings, we conducted additional experiments on GPQA, MedQA, and IFEval, which include biology, medical, and physics question-answering tasks, as well as instruction-following tasks in open-domain settings. As shown in Table 3, the results indicate that the collaboration frequency consistently remains below 20% across various methods, tasks, and model combinations. Furthermore, we observe a decreasing trend in collaboration frequency as the ratio of model parameters decreases.

We also found that the collaboration rate of general models on domain-specific tasks is slightly higher than on general tasks. When extending model collaborations from generalist to specialist tasks, we anticipate that the collaboration frequency will decrease due to the narrower distribution of domain-specific terminology. However, the limited availability of comprehensive specialized model series constrains further analysis at this stage, leaving this exploration for future work.

Table 3: Collaboration frequency for Qwen models across GPQA, IFEval, and MedQA datasets.

| $CoF_{\text{lower}}$ | GPQA | | | IFEval | | | MedQA | | |
|---|---|---|---|---|---|---|---|---|---|
| | SD | CD | PT | SD | CD | PT | SD | CD | PT |
| Qwen1.5-0.5B w 7B | 0.162 | 0.211 | 0.157 | 0.208 | 0.298 | 0.200 | 0.230 | 0.296 | 0.225 |
| Qwen1.5-1.8B w 7B | 0.130 | 0.198 | 0.133 | 0.174 | 0.238 | 0.164 | 0.194 | 0.314 | 0.190 |
| Qwen1.5-4B w 7B | 0.099 | 0.155 | 0.098 | 0.149 | 0.221 | 0.145 | 0.169 | 0.308 | 0.165 |

We present the fitting line for the parameter scale ratio law of the Pythia series in Figure 12. In comparison to the Qwen series and OpenELM series shown in Figures 4 and 5, the Pythia series exhibit weaker performance, making it more challenging to fit the lines accurately.

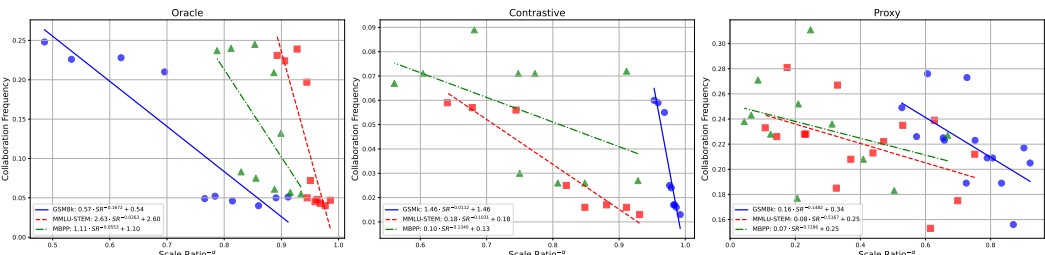

Figure 12: The relationship between models' parameters ratios and the lower bounds of collaboration frequency (**Pythia Series**).

## E.2 RESULTS ON COLLABORATION POSITION

We provide heatmaps of mismatch positions relative to Pythia model combinations for contrastive decoding and emulator fine-tuning in Figures 14 and 15 respectively. These heatmaps reveal findings similar to those obtained from oracle decoding and support the adage that "well begun is half done".

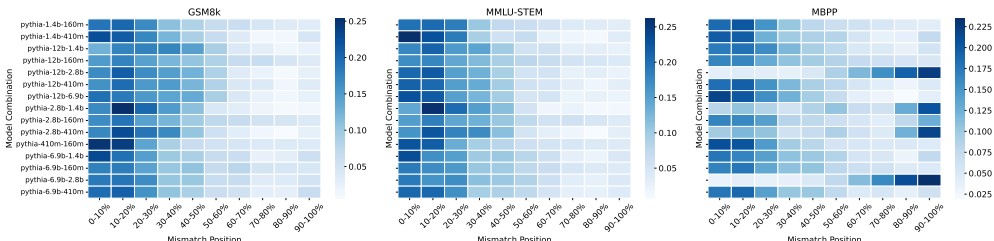

Figure 13: Percentage of mismatch positions relative to length for model combinations across various datasets (**Pythia Series, Oracle Decoding**).

The difference between Figures 10 and 11 lies in the selection of tokens within the generated content. Specifically, we select only one token for the former and five continuous tokens for the latter. This relationship between the top-k logits for each position and mismatches with large models is clearly evident in the visualization. The results further indicate that identifying mismatch positions benefits from additional context at each step.

We provide additional model combinations for visualization of token-level uncertainty, as shown in Figure 19, 20, 21, 22, 23, 24. This figure also demonstrates that the uncertainty in tokens of SLMs results in incorrect answers. Examples illustrating uncertainty in SLMs using contrastive decoding and proxy tuning are presented in Figures 25 and 26, respectively.

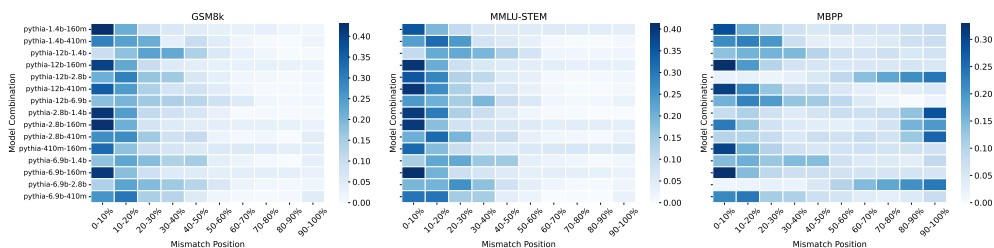

Figure 14: Percentage of mismatch positions relative to length for model combinations across various datasets (**Pythia Series, Contrastive Decoding**).

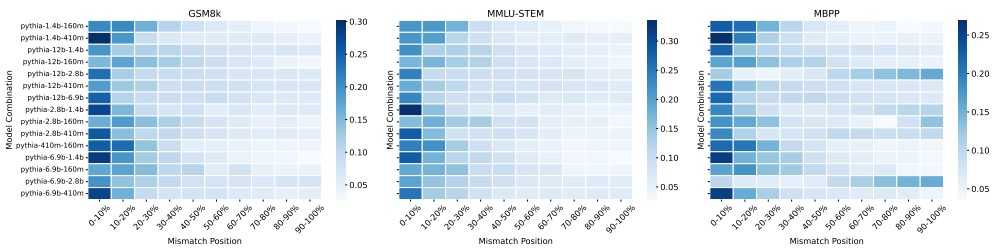

Figure 15: Percentage of mismatch positions relative to length for model combinations across various datasets (**Pythia Series, Proxy FineTuning**).

# F    PRACTICAL APPLICATION

## F.1    QUANTITIVE RESULTS OF UNCERTAINTY ANALYSIS

| Models | Metric | GSM8k | | MMLU | | MBPP | |
|--------|--------|---------|---------|---------|---------|---------|---------|
| | | 5 tokens | 1 token | 5 tokens | 1 token | 5 tokens | 1 token |
| Qwen1.5 | SC | 0.465 | 0.503 | 0.445 | 0.457 | 0.470 | 0.469 |
| | DBI | 0.806 | 0.805 | 0.838 | 0.917 | 0.772 | 0.909 |
| | MCCD | 7.533 | 18.176 | 11.036 | 15.64 | 13.431 | 16.156 |
| Pythia | SC | 0.465 | 0.358 | 0.485 | 0.286 | 0.464 | 0.315 |
| | DBI | 0.790 | 1.180 | 0.755 | 1.416 | 0.779 | 1.300 |
| | MCCD | 21.584 | 14.125 | 22.584 | 16.289 | 21.325 | 16.843 |

Table 4: Correlation Between Match/Mismatch Tokens and Top-K Token Logits of SLMs.

To strengthen the evidence supporting our uncertainty analysis, we provide the corresponding quantitative results from the clustering in Figures 10 and 16, generalizing across all model combinations and methods. We utilize the following metrics to evaluate the correlation between matched and mismatched token logits:

- Silhouette Coefficient (SC). This metric (range: -1 to 1) assesses clustering quality by comparing intra-cluster cohesion and inter-cluster separation. Values > 0.5 indicate strong clustering performance. A high SC value derived from Pearson or Spearman correlation demonstrates that the metric aligns well with the data.

- Davies-Bouldin Index (DBI). The DBI (range: $[0, \infty]$) measures clustering compactness and separation, where lower values (<1) suggest better clustering quality. A low DBI derived from correlation methods indicates effective uncertainty estimation.

- Mean Cluster Center Distance (MCCD). MCCD measures the separation between cluster centers, with larger values indicating better distinction. Correlation methods that amplify these distances demonstrate their alignment with the data.

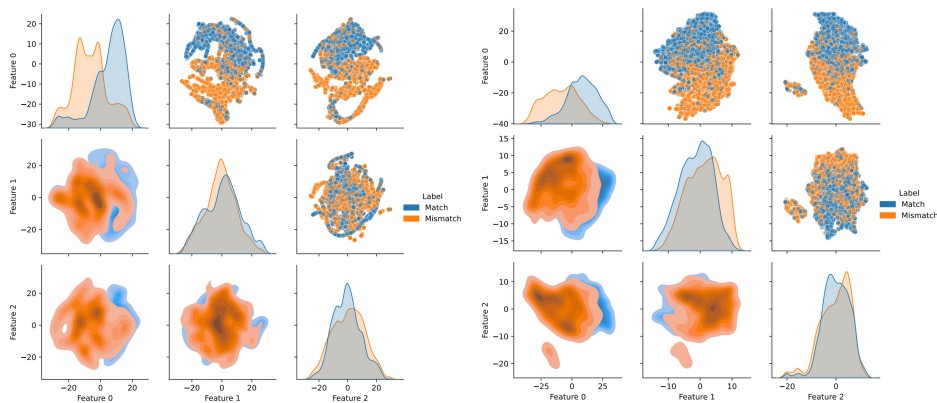

Figure 16: KDE and scatter visualization of feature distribution for mismatch/uncertain tokens (1 & 5 tokens) (**Pythia Series**).

As shown in Table 4, our results demonstrate the effectiveness of uncertainty estimation: SC values are consistently close to 0.5; DBI values are below 1, indicating compact and well-separated clusters; MCCD values range between 10 and 20, reflecting robust inter-cluster distinction. An exception is observed with Pythia series models, likely due to their insufficient pretraining. In conclusion, these results validate the effectiveness of Finding 4 discussed in Section 5.2.2.

## F.2 TOKEN-LEVEL ROUTING FOR COLLABORATIVE DECODING

Based on the quantitative results presented in Section F.1, we propose practical applications for proxy tuning, illustrated in Figure 11 and Figure 17. By utilizing only a few key positions for collaborative decoding, we can achieve higher accuracy outputs based on SLMs. These results highlight the effectiveness of token-level routing from SLMs to mix-scaled models using only the logits of the top-1 token from SLMs. Additionally, we provide a qualitative analysis of other related metrics, such as entropy and perplexity of the next token from SLMs, in Figure 18. These metrics exhibit similarly distinguishable distributions and offer preliminary potential for routing.

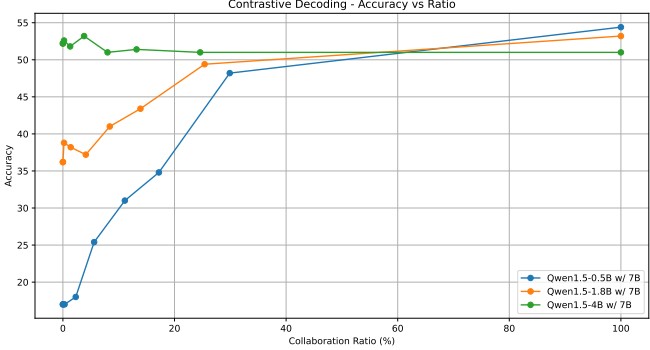

Figure 17: Routing with Top-1 Token Logits of SLM for Contrastive Decoding (**Qwen Series**).

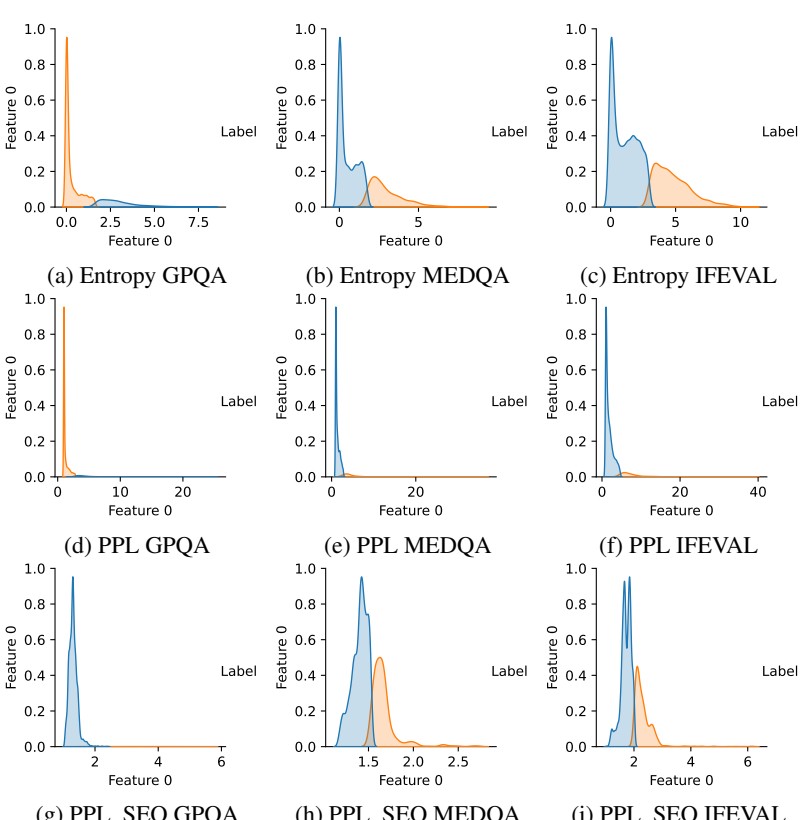

Figure 18: Comparison of Routing Metrics across Datasets

**Question:** The expenditure of Joseph in May was $500. In June, his expenditure was $60 less. How much was his total expenditure for those two months? Let's think step by step

**LLM:** Qwen-1.8B

**SLM:** Qwen-0.5B

**SLM Results(Correct):** In June, Joseph's expenditure was $500 - $60 = $440.\nSo in total, his total expenditure for May and June was $500 + $440 = $940.\nThe answer is 940

**Collaborative Decoding Results(Wrong):**

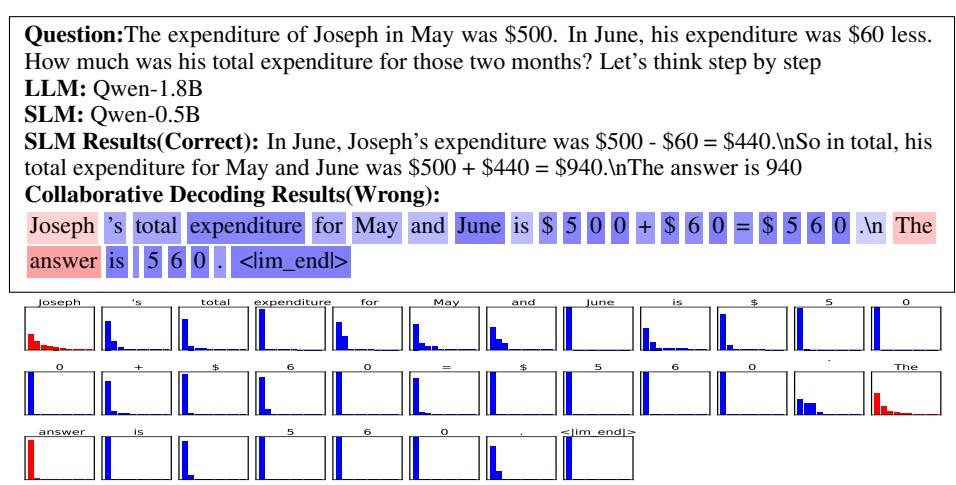

Figure 19: Visualization of Token Contributions. Red signifies mismatch tokens, Blue indicates match tokens, with darker shades representing higher weights.

**Question:** Grace weighs 125 pounds. Alex weighs 2 pounds less than 4 times what Grace weighs. What are their combined weights in pounds? Let's think step by step

**LLM:** Qwen-1.8B

**SLM:** Qwen-0.5B

**SLM Results(Wrong):** Alex weighs 4 * 125 - 2 = 480 pounds.\nGrace weighs 125 + 480 = 625 pounds.\nTheir combined weights in pounds are 625 + 125 = 750 pounds.\n\nTherefore, the answer is 750.

**Collaborative Decoding Results(Wrong):**

Alex weighs 4 * 1 2 5 - 2 = 5 0 2 pounds .\n Their combined weight is 1 2 5 + 5 0 2 = 6 2 7 pounds .\n The answer is 6 2 7 . <|im_end|>

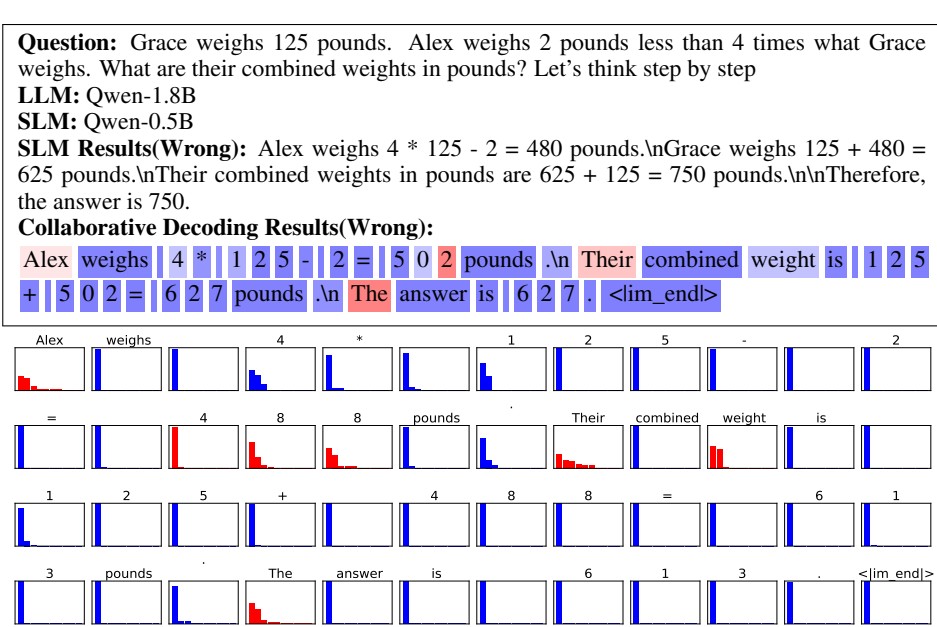

Figure 20: Visualization of Token Contributions. Red signifies mismatch tokens, Blue indicates match tokens, with darker shades representing higher weights.

**Question:** A bakery has 40 less than seven times as many loaves of bread as Sam had last Friday. If Sam had seventy loaves of bread last Friday, how many loaves of bread does the bakery have? Let's think step by step

**LLM:** Qwen-1.8B

**SLM:** Qwen-0.5B

**SLM Results(Wrong):** The bakery has 7 * 40 = 280 loaves of bread.\nIf Sam had 70 loaves of bread last Friday, then the bakery has 280 - 70 = 210 loaves of bread.\nThe answer is 210

**Collaborative Decoding Results(Wrong):**

If Sam had 7 0 lo aves of bread last Friday , then the bakery has 7 0 - 4 0 = 3 0 lo aves of bread .\n #### 3 0 \n The answer is : 3 0 <|im_end|>

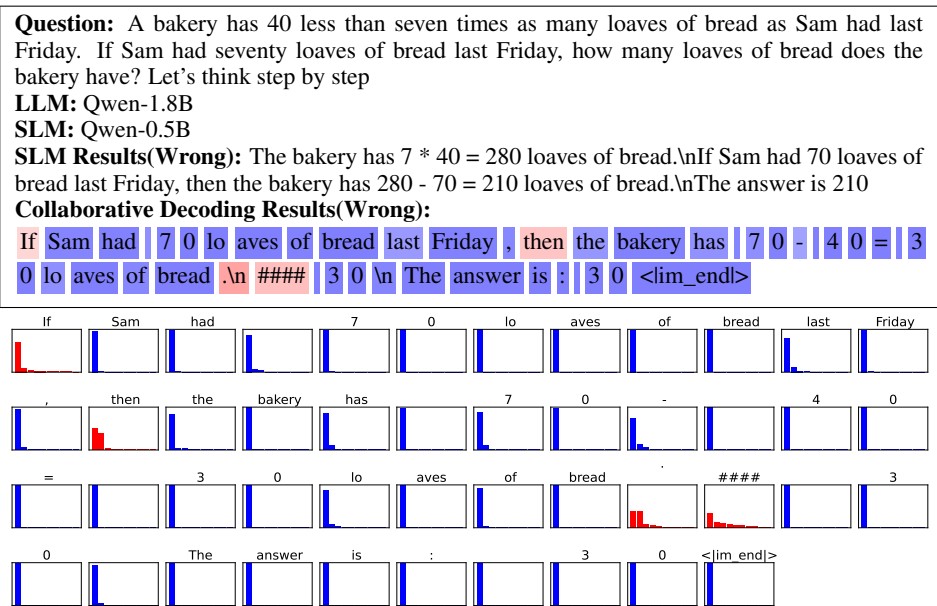

Figure 21: Visualization of Token Contributions. Red signifies mismatch tokens, Blue indicates match tokens, with darker shades representing higher weights.

**Question:** The expenditure of Joseph in May was $500. In June, his expenditure was $60 less. How much was his total expenditure for those two months? Let's think step by step

**SLM Results(Correct):** In June, Joseph's expenditure was $500 - $60 = $440.\nSo in total, his total expenditure for May and June was $500 + $440 = $940.\nThe answer is 940

**LLM:** Qwen-4B

**SLM:** Qwen-1.8B

**Collaborative Decoding Results(Correct):**

In May , Joseph spent $ 5 0 0 .\n In June , his expenditure was $ 6 0 less , so he spent $ 5 0 0 - $ 6 0 = $ 4 4 0 .\n To find the total expenditure for those two months , we add the expenditures : $ 5 0 0 + $ 4 4 0 = $ 9 4 0 \n The answer is 9 4 0 . <|im_end|>

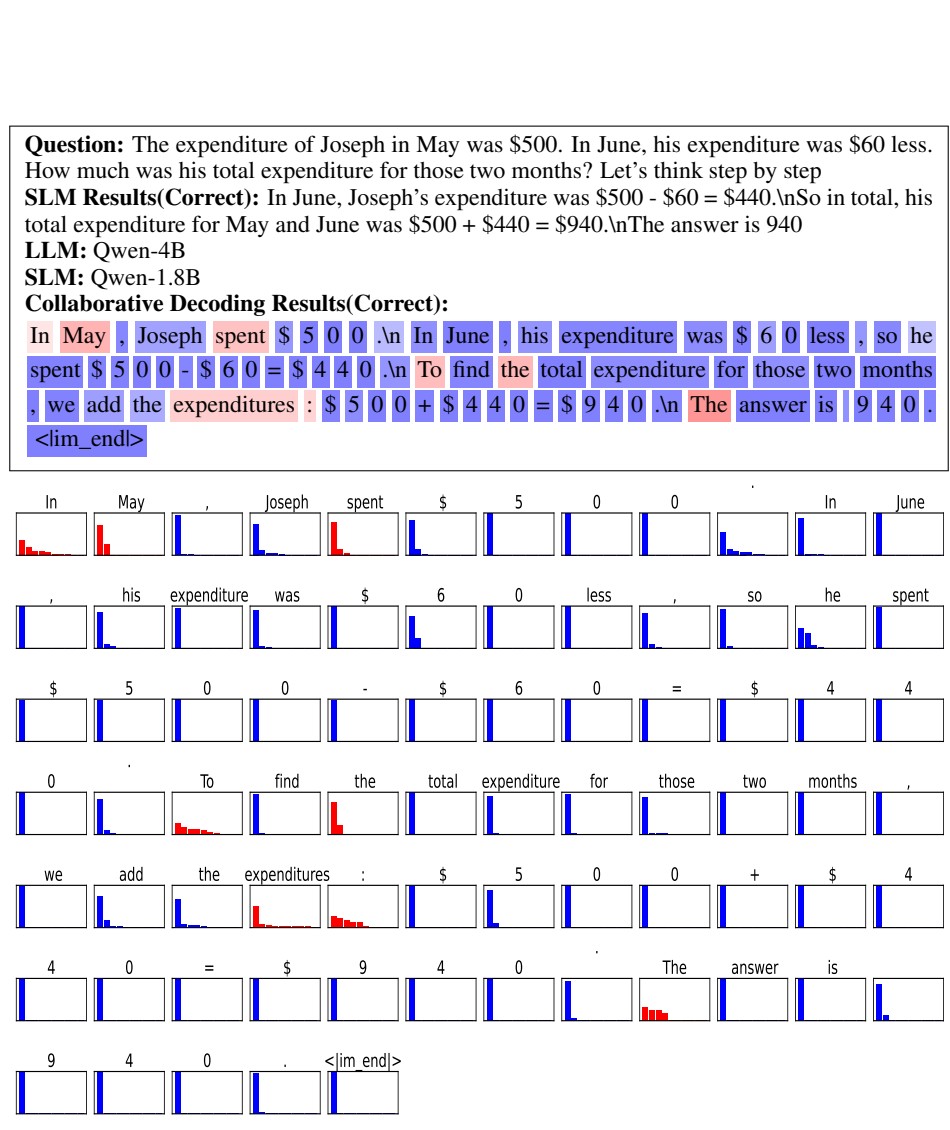

Figure 22: Visualization of Token Contributions. Red signifies mismatch tokens, Blue indicates match tokens, with darker shades representing higher weights.

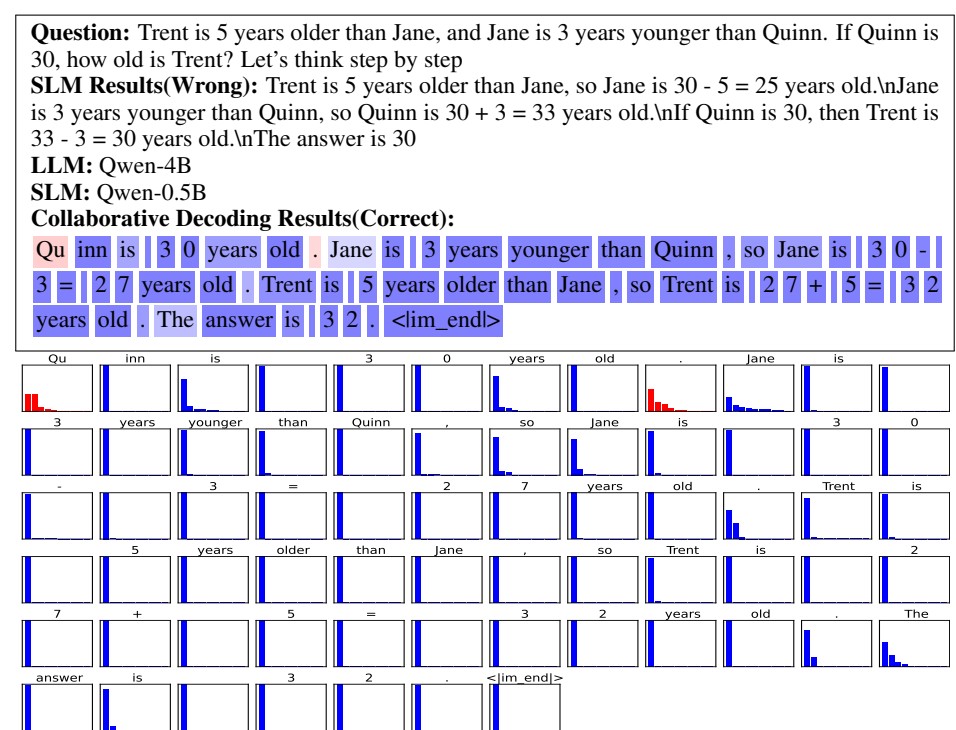

Figure 23: Visualization of Token Contributions. Red signifies mismatch tokens, Blue indicates match tokens, with darker shades representing higher weights.

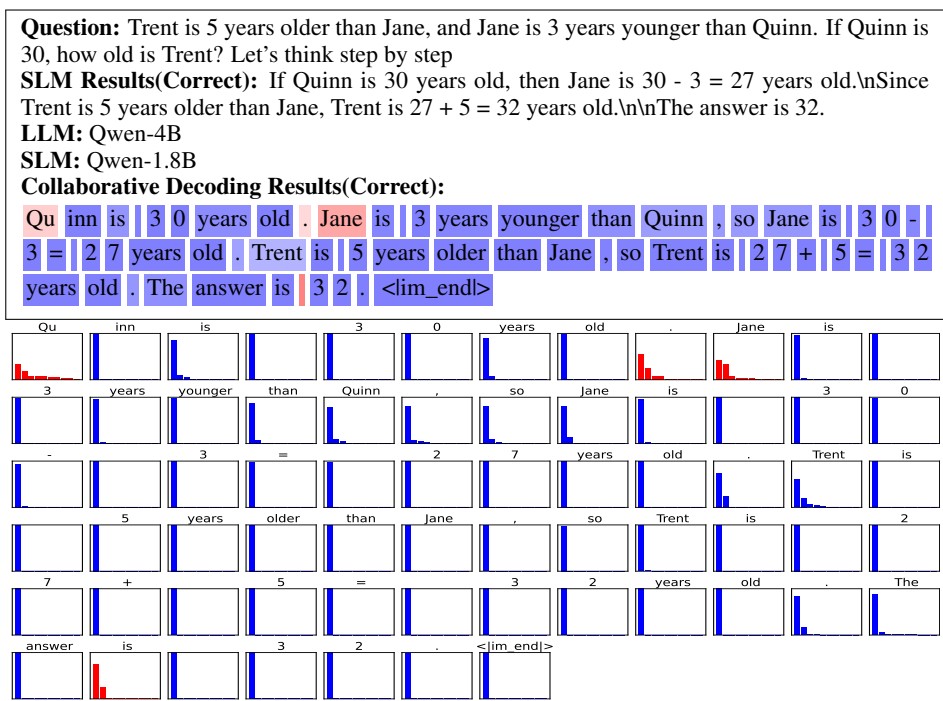

Figure 24: Visualization of Token Contributions. Red signifies mismatch tokens, Blue indicates match tokens, with darker shades representing higher weights.

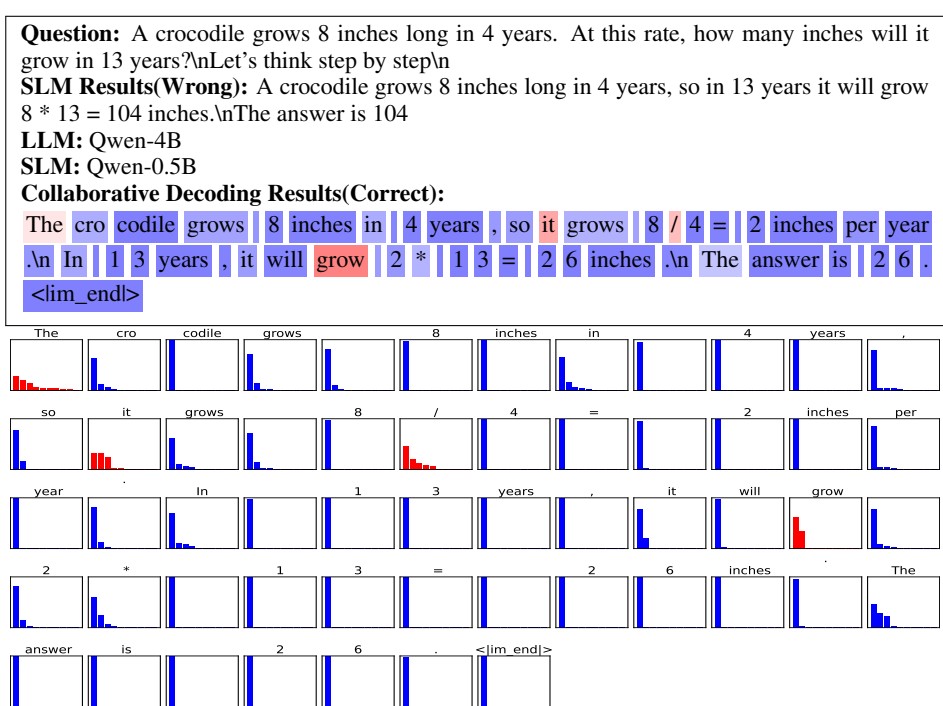

Figure 25: Visualization of Token Contributions in **Contrastive Decoding**. Red signifies mismatch tokens, Blue indicates match tokens, with darker shades representing higher weights.

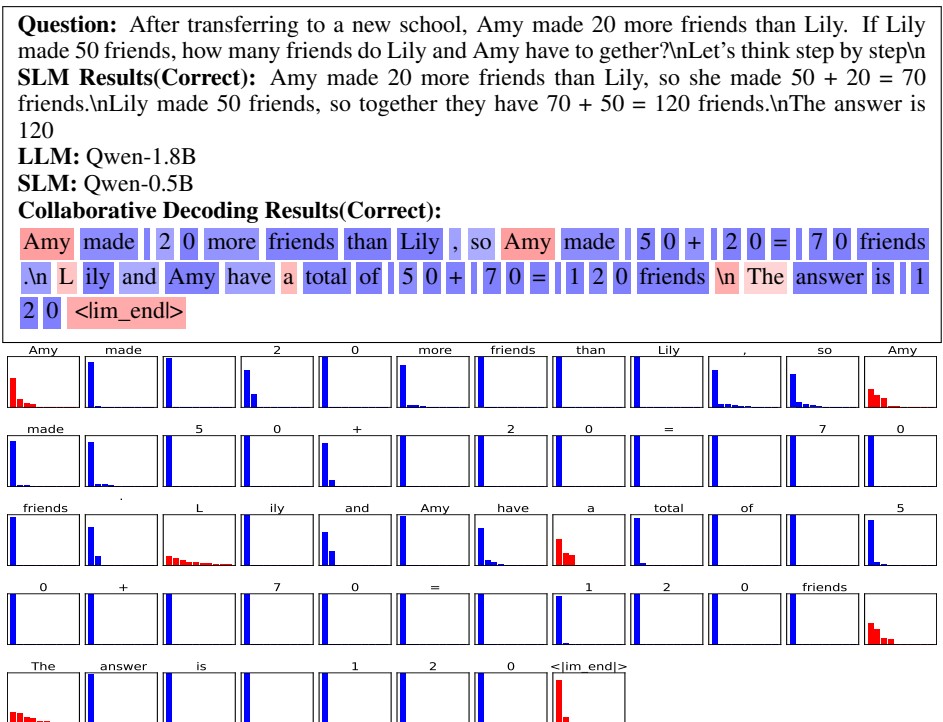

Figure 26: Visualization of Token Contributions in **Proxy Tuning**. Red signifies mismatch tokens, Blue indicates match tokens, with darker shades representing higher weights.

