# OpenReview forum: "Fast and Slow Generating: An Empirical Study on Large and Small Language Models Collaborative Decoding"
_ICLR.cc/2025/Conference — Submitted to ICLR 2025_

### Official Review · Reviewer_VTMo · 2024-11-01

**Soundness:** 1
**Presentation:** 1
**Contribution:** 2
**Rating:** 3
**Confidence:** 3

**Summary:**

This paper presents an investigation into collaborative decoding between small and large language models, attempting to formalize it from the perspective of a system 1 / system 2 collaboration, where system 1 operates quicly and intuitively, while system 2 functions in a more slow and deliberate manner. The paper focuses on the differences between system 1 and 2 in the context of decoding, when system 1 would underperform compared to system 2 and how efficiency of the compound system can be improved. For their investigation, the authors use the Qwen and Pythia series. To evaluate the system, they consider MMLU-STEM, GSM8k and MBPP. The analysis focusses on two aspects of collaboration: frequency and position, where the former refers to how often the models should interact, where as the second one refers to the specific points of interaction. They find thta collaborative interactions are most critical at the beginning of the generation, and that the optimal frequency is around
 80-20, depending on the task.

**Strengths:**

The paper asks an interesting question and presents several findings. The idea to take inspiration from system 1 and system 2 is interesting.

**Weaknesses:**

My main qualm with the work is the presentation of the paper, which almost reads like a slide deck: plenty of conclusions and graphics, but little to no details about how the experiments are actually set up or how the conclusions are drawn. I also don't see any evidence of how well the collaborative decoding actually works (that is, there are no accuracy scores reported), and how that may depend on the frequency or place of collaboration). The many figures are hardly described. There is also no discussion of how the results are different between the benchmarks and whether that may make sense given the topics.

Lastly, while I like the idea of interpreting collaborative decoding as a system-1 system-2 scenario, but the current work does not really convince me that it makes sense to explore collaborative decoding with SLMs and LLMs in this way. Wouldn't LLMs be better both at the intuition and the deliberate reasoning?

In sum, it could be that the paper contains many interesting results, but if so, the current presentation does not do them justice.

NB: the related work section is in the appendix and is not even referred to

**Questions:**

Could you elaborate on the motivation of using system 1 - system 2 reasoning for collaborative decoding with SLMs and LLMs, specifically?

---

> ### Author Response · Authors · 2024-11-22
> **Response - 1**
>
> We sincerely appreciate the time and effort you have invested in reviewing our paper. We apologize for any confusion and would like to clarify our motivations and experimental settings.
>
> ### Q1: About the Presentation of the Paper
> Thank you for your comments. We would like to share our thoughts on the “slide deck” presentation style, which aims to make the conclusions clearer, as **Reviewer DHMi** aptly summarized.
> Given the page limitations, we chose to highlight the main analyses and conclusions for each figure and finding. However, we recognize that this approach may impose a burden of understanding, especially for readers less familiar with the field.
> To address this:
> - We will strengthen the connections between the figures and their corresponding conclusions in response to the questions raised. Preliminary connection instructions will be provided in #Q2.
> - We apologize for the lack of detailed descriptions of the experiments and the process of deriving conclusions. In the revised version, we will reorganize the paper to include related work and preliminary knowledge in the main text.
>
> ### Q2: About working mode of collaborative decoding
> #### A1: Running Example
> Our primary objective is to analyze the frequency of collaboration in various decoding settings.
> In our research, we explore collaborative decoding (CoDec) at all steps ($CoF=1$), for example:
> ```
> [User]: "Who is Donald Trump?"
> [Assistant]: "Donald Trump is the former President of the United States, who is 78 years old now."
> ```
>
> For a lower collaboration frequency ($CoF_{\text{lower}}$), we input the outputs of CoDec into smaller models token by token to assess the consistency of top tokens. (CoDec represents speculative decoding ,contrastive decoding or proxy tuning)
> ```
> - First token verification, match:
> 	- CoDec: [Assistant]: "Donald"
> 	- Small: [Assistant]: "Donald"
> - Second token verification, match:
> 	- CoDec: [Assistant]: "Donald Trump"
> 	- Small: [Assistant]: "Donald Trump"
> - ...
> - 5th token verification, mismatch:
> 	- CoDec: [Assistant]: "Donald Trump is the former President"
> 	- Small: [Assistant]: "Donald Trump is President"
> - ... (match)
> - 14th token verification, mismatch:
> 	- CoDec: [Assistant]: "Donald Trump is the former President of the United States, who is 78"
> 	- Small: [Assistant]: "Donald Trump is the former President of the United States, who is 80"
> - ... (match)
> ```
>
> Assuming there are three mismatched tokens (e.g., "former", "78"), the calculated $CoF_{\text{lower}}=\frac{2}{18}$. However, unnecessary collaborations may occur even when matches are identified, leading to an variable where $CoF_{\text{lower}} \leq CoF \leq 1$. This motivates our investigation into the lower bounds of collaboration frequency, aiming to achieve similar outputs as full collaborative decoding with minimal collaborative steps. Our findings demonstrate this is a universal phenomenon across different collaborative decoding methods.
>
> Speculative decoding currently selects a fixed number of tokens (K-tokens) for generation-verification, which does not effectively reach $CoF_{\text{lower}}$. In contrast, methods such as contrastive decoding and proxy tuning entail collaborations at each step ($CoF=1$), which may not always be necessary.
>
> #### A2: Raw Accuracy Scores of Different Methods
> While accuracy scores are not the primary focus of our experiments—where the outputs of mix-scaled models are treated as golden outputs—we provide raw accuracy scores for different collaborative methods to support the validity of our approach.
>
> The results demonstrate the effectiveness of collaborative decoding, showing that mix-scaled models outperform small models operating independently. Furthermore, these findings underscore the potential to optimize collaboration efficiency based on the insights presented in our paper.
>
> - Table 1: Accuracy of Different Collaborative Decoding Methods on GSM8k
>
> | Model        | Method | Qwen1.5-0.5B | Qwen1.5-1.8B | Qwen1.5-4B  | Qwen1.5-7B  |
> | ------------ | ------ | ------------ | ------------ | ----------- | ----------- |
> | Qwen1.5-0.5B | Self   | 17.0 (Self)  | -            | -           | -           |
> | Qwen1.5-1.8B | SD     | 36.2         | 36.2 (Self)  | -           | -           |
> |              | CD     | 33.4         | \            | -           | -           |
> |              | PT     | 38.0         | \            | -           | -           |
> | Qwen1.5-4B   | SD     | 52.2         | 52.2         | 52.2 (Self) | -           |
> |              | CD     | 48.8         | 47.0         | \           | -           |
> |              | PT     | 49.8         | 51.2         | \           | -           |
> | Qwen1.5-7B   | SD     | 57.0         | 57.0         | 57.0        | 57.0 (Self) |
> |              | CD     | 54.4         | 53.2         | 51.0        | \           |
> |              | PT     | 57.0         | 57.0         | 56.8        | \           |

---

> ### Author Response · Authors · 2024-11-22
> **Response - 2**
>
> ### Q3: Discussion of the Figures (Results of Various Methods on Benchmarks)
>
> #### A1: Connections between Figures and Conclusions
> We provide detailed connections between the results of various methods on benchmarks presented in the figures and our final findings. Specifically, we conducted experiments on three methods (**speculative decoding, contrastive decoding, proxy tuning**) across three benchmarks (**GSM8k, MMLU, MBPP**) and two model series (**Qwen, Pythia**) to derive our four main findings.
>
> The results for these combinations sufficiently support each finding, as outlined below:
>
> - **Finding 1: 20% Collaborations — The 2:8 Law**
> 	- **Figures 2 (Qwen) and 3 (Pythia):** These figures show that the average collaboration frequency (interface rate) across different tasks is consistently less than 20%. Among the methods, contrastive decoding achieves much lower collaboration frequencies than speculative decoding and proxy tuning.
> 	- **Task Variability:** While collaboration frequency varies across tasks and model series due to differing model capabilities, the values remain approximately 20%.
> 	- **Model Combinations:** Collaboration frequency decreases as the parameter ratio between large and small models decreases. This implies that models with similar capabilities require less frequent collaboration, which also contributes to **Finding 2**.
> - **Finding 2: Parameters Scale Ratio Law**
> 	- **Figure 4 (Qwen):** This figure presents the fitting line for the parameter scale ratio between large and small models, showing a strong correlation with lower collaboration frequencies. The fitting effect demonstrates the validity of our parameter scale ratio law.
> 	- **Task and Method Variability:** While the quality of the fitting line varies by task and method, the conclusions remain consistent.
> 	- Figure 5 (Pythia) highlights how the fitting effect is influenced by model underperformance. To strengthen this finding, we provide additional results from OpenELM, which exhibit low fitting errors and further support the effectiveness of our scale ratio law.
> - **Finding 3: Well Begun is Half Done**
> 	- **Figures 6 (Qwen) and 7 (Pythia):** These figures illustrate the mismatch rate across the generated sequence. They reveal that most mismatches occur at the **beginning** of the sequence. As generation progresses, SLMs increasingly align with mix-scaled models, leveraging the shared context.
> 	- **Task Variability:** The percentage of mismatches in a sequence varies by task, influenced by task difficulty and model capability.
> - **Finding 4: Last in Uncertain Tokens**
> 	- **Figure 10:** This figure compares the logits of each token generated by SLMs to those of mix-scaled models, showing that mismatches predominantly occur in high-entropy positions (indicating high uncertainty).
> 	- **Figures 11, 12, 13:** These figures analyze the top- token logits at each step, clustering tokens into match and mismatch categories. The results reveal a strong correlation between uncertainty (high-entropy positions) and matching labels.
> 	- To ensure robustness, we compute average correlation scores across all tasks and methods. This finding identifies key positions for collaboration during decoding in SLMs, contributing to performance-cost optimization.

---

> ### Author Response · Authors · 2024-11-22
> **Response - 3**
>
> #### A2: Additional results of new tasks and models
> To further support our findings, we conducted additional experiments on new tasks (**GPQA, IFEval, MedQA**) and a new model series (**OpenELM**). The results validate the generalizability of our conclusions, as outlined below:
>
> - Table 2: Results of  $CoF_{\text{lower}}$  on Additional Domain Tasks
> 	- The results indicate that  $CoF_{\text{lower}}$  is consistently below 20% across various methods, tasks, and model combinations. Furthermore, we observe a decreasing trend in  $CoF_{\text{lower}}$  as the ratio of model parameters decreases.
> 	- We also found that the collaboration rate of general models on domain tasks is slightly higher than that on general tasks.
>
> | Task               | GPQA  |       |       | IFEval |       |       | MedQA |       |       |
> | ------------------ | ----- | ----- | ----- | ------ | ----- | ----- | ----- | ----- | ----- |
> | Method / CoF_lower | SD    | CD    | PT    | SD     | CD    | PT    | SD    | CD    | PT    |
> | Qwen1.5-0.5B w/ 7B | 0.162 | 0.211 | 0.157 | 0.208  | 0.298 | 0.2   | 0.23  | 0.296 | 0.225 |
> | Qwen1.5-1.8B w/ 7B | 0.13  | 0.198 | 0.133 | 0.174  | 0.238 | 0.164 | 0.194 | 0.314 | 0.19  |
> | Qwen1.5-4B w/ 7B   | 0.099 | 0.155 | 0.098 | 0.149  | 0.221 | 0.145 | 0.169 | 0.308 | 0.165 |
>
> - Table 3: Line Fitting Results of OpenELM Models
> 	- Given the following formula of scale ratio law $$CoF_{\text{lower}}=\gamma \cdot {R}^{-\alpha}+\beta$$where $R=\frac{N_l}{N_s}$, we compute the coefficients with model parameters and collaboration frequency.
> 	- This table presents the line fitting results for oracle decoding, including fitting error, fitting coefficients, and the final  x (i.e., $R^{-\alpha}$) and  y  values on the fitting curve.
> 	- The results demonstrate a strong fitting effect, confirming the generalizability of our findings across different model families.
> 	- These results further indicate that the performance of collaborative decoding is influenced by the underlying model’s performance.
>
> | Task      | Formula & Fitting Error                     | Coordinates | 270M/450M | 450M/1.1B | 1.1B/3B | 270M/1.1B | 450M/3B | 270M/3B |
> | --------- | ------------------------------------------- | ----------- | --------- | --------- | ------- | --------- | ------- | ------- |
> |           |                                             | Ratio       | ≈1.67     | ≈2.44     | ≈2.73   | ≈4.07     | ≈6.67   | ≈11.11  |
> | GSM8k     | $\gamma=4.66, \alpha=-0.0041, \beta=4.67$   | X axis      | 0.9979    | 0.9964    | 0.9959  | 0.9943    | 0.9923  | 0.9902  |
> |           | MSE Loss = 1.16e-6                          | Y axis      | 0.0250    | 0.0320    | 0.0320  | 0.0420    | 0.0490  | 0.0610  |
> | MMLU-STEM | $\gamma=5.02, \alpha=-0.0022, \beta=5.04$   | X axis      | 0.9989    | 0.9981    | 0.9978  | 0.9969    | 0.9959  | 0.9948  |
> |           | MSE Loss = 2.25e-6                          | Y axis      | 0.0280    | 0.0300    | 0.0350  | 0.0350    | 0.0420  | 0.0490  |
> | MBPP      | $\gamma=23.83, \alpha=-0.0007, \beta=23.84$ | X axis      | 0.9996    | 0.9994    | 0.9993  | 0.9990    | 0.9987  | 0.9983  |
> |           | MSE Loss = 7.04e-5                          | Y axis      | 0.0220    | 0.0170    | 0.0380  | 0.0200    | 0.0480  | 0.0500  |

---

> ### Author Response · Authors · 2024-11-22
> **Response - 4**
>
> ### Q4: Motivation and Analogy of System 1 and 2 with Collaborative Decoding
>
> In this work, we draw inspiration from the analogy of System 1 and System 2, simplifying their collaboration into **Fast and Slow thinking**. System 1 efficiently handles approximately 95% of routine tasks, while System 2 is reserved for deliberately addressing the remaining 5% of complex work [1]. Together, they demonstrate the power of **high-efficiency collaboration**.
>
> We adopt this high-efficiency motivation to model the collaborative decoding methods between **fast and slow** (or **small and large**) models. Our experimental findings (Findings 1 and 2) show that small, fast models generate approximately 80% of tokens during the answering process, while large, slow models contribute the remaining 20%.
>
> Looking forward, we aim to expand these collaborative mechanisms to **reasoner and knowledger models**, such as OpenAI’s o1 and GPT-4, which not only embody the fast/slow model paradigm but also represent intuitive and deliberate thinking. Preliminary experiments reinforce our findings, indicating successful collaboration between o1 and existing large language models.
>
> [1] Booch, Grady, et al. "Thinking fast and slow in AI." Proceedings of the AAAI Conference on Artificial Intelligence. Vol. 35. No. 17. 2021.
>
> ### Q5: Position of Related Works in the Paper
>
> We appreciate your feedback regarding the placement of related works. To improve clarity, we will reorganize the structure of the paper. Specifically:
> - **Revised structure:** We will simplify the presentation of the main text to enhance readability.
> - **Related works:** We will introduce related work briefly in the main text, ensuring it is more integrated and accessible.
>
> We hope these responses address your concerns and provide further clarity. Thank you once again for your constructive feedback and valuable suggestions, which have been instrumental in improving our work.

---

> ### Comment · Reviewer_VTMo · 2024-11-22
>
> Dear authors,
>
> Thank you very much for your explanations. I remain with my earlier comment that the questions asked in your article are interesting and that there are likely many valuable results in your paper. I appreciate that you, among other things, acknowledge in your response that it is necessary to strengthen the connections between the figures in the corresponding conclusions (and give some of those explanations in your rebuttal), that you agree that experimental details are needed for readers to validate your experiments and that the paper could benefit from some more explicit reasoning an how the conclusions were arrived.  I believe that doing all these changes would drastically improve your paper (though I would have to take another few hours to re-review to make sure that the conclusions make sense given the added experimental details, descriptions, motivation and reasoning). I also think that these changes would be quite substantial (as confirmed also by the length of your response) and -- as I said in the line before -- reviewing them would take almost as much time as reviewing the paper in the first place. For me, this is beyond the expectations of a rebuttal phase and I will thus stick with my recommendation to reject your work. Of course, it is just a one man's opinion, and I want also to acknowledge that I do think your paper has promise and carries some interesting ideas. The other reviewers appear to be more positive about this work than myself, so perhaps the AC will just overrule my specific opinion :). I hope in any case that my comments were useful to improve your work.

---

> > ### Author Response · Authors · 2024-11-23
> > **Clarifying Misunderstandings**
> >
> > Thank you for your response. We noticed there are still significant misunderstandings regarding both our response and the paper. We’d like to clarify that **our detailed response and additional results do not introduce “substantial changes” beyond what was presented in the original submission**. Below, we address each point to provide further explanation:
> >
> > - **Clarifying System 1 & 2 Analogy and Collaborative Decoding Setup:**
> > 	- A significant portion of our response is dedicated to helping the reviewer better understand the analogy of System 1 and System 2 **(Q4)** and the working mechanism of collaborative decoding **(Q2-A1)**. These aspects are already highlighted in our paper. Specifically:
> > 		- The motivation for high-efficiency collaboration is discussed in the Introduction (**Lines 084-098**).
> > 		- The operational details and experimental setups can be inferred from the related works (e.g., speculative decoding[1], contrastive decoding[2], and proxy tuning[3]) and our explanations in **Sections 3.1, 3.2, and Appendix C**.
> > 	- These ensure that our results are reproducible based on the information provided. Additionally, we open-source our code within a unified framework in [Anonymous Repository](https://anonymous.4open.science/r/ICLR2025_Anonymous-127D) for reference.
> > - **Additional Results and Their Relevance:**
> > 	- The remaining part of our response provides additional results **(Q2-A2, Q3-A2)** to demonstrate the generalizability of our findings across more tasks and models. As per the [ICLR Reviewer Guide](https://iclr.cc/Conferences/2025/ReviewerGuide), these supplementary experiments do not alter the conclusions of our submission but instead validate the existing results more thoroughly.
> > - **Clarifying Our Focus:**
> > 	- Once the setup is clear, it becomes evident that our study focuses on identifying common features of various collaborative decoding methods. Previous works [1,2,3] have already demonstrated the performance of collaborative decoding.
> > 	- In our study, the outputs of collaborative decoding are considered as ground truth (or “golden” outputs). **Therefore, accuracy results for experiments are not the primary goal.** Instead, we aim to explore the minimal frequency and key positions of collaboration, particularly from the perspective of smaller models.
> > 	- Based on the visualization, we can clearly derive the common findings across various benchmarks and models, also as highlighted by Reviewer DHMi. While there may be subtle differences in the results due to variations in tasks and the capabilities of different models, these do not affect our primary findings **(Q3-A1)**. We will include a separate section to discuss these nuances.
> >
> > In conclusion, we believe **our submission provides sufficient experimental details for reproducibility, and our response does not introduce significant changes beyond the original paper.**
> >
> > We are happy to engage in further discussions to address any remaining points of confusion.
> >
> > [1] Leviathan, Yaniv, Matan Kalman, and Yossi Matias. "Fast inference from transformers via speculative decoding." _International Conference on Machine Learning_. PMLR, 2023.
> >
> > [2] Li, Xiang Lisa, et al. "Contrastive decoding: Open-ended text generation as optimization." arXiv preprint arXiv:2210.15097 (2022).
> >
> > [3] Liu, Alisa, et al. "Tuning language models by proxy." _arXiv preprint arXiv:2401.08565_ (2024).

---

> > > ### Comment · Reviewer_VTMo · 2024-11-25
> > >
> > > Dear authors,
> > >
> > > I see. I had understood from your answers that you agreed with my suggestions and were open to incorporating them, and I am sorry to hear that is not the case. I really don't intend to be mean about this, and I definitely see promise in your ideas, but I still believe the paper really needs substantial revisions to be ready for publication, and the *very* lengthy responses (with much information not contained in the paper), do not change my mind about this. I read the reviews of reviewer zvri and DHMi, who give higher scores, but their reviews are very short with not much info to go on. In other words, for me this paper is simply not ready for publication. I will not lower my score, but I will not make it higher either, I am sorry.

---

### Official Review · Reviewer_gboG · 2024-11-04

**Soundness:** 3
**Presentation:** 2
**Contribution:** 2
**Rating:** 6
**Confidence:** 4

**Summary:**

The paper analyzes the patterns of collaboration between SLMs and LLMs when used in a collaborative decoding/training setup. By analyzing this behavior across multiple collaboration setups, tasks, and model families, the authors draw the following conclusions:
- The collaboration frequency peaks at about 20%, with the maximum collaboration happening when there's the biggest gap in the model sizes. In fact, there's an inverse scaling law connecting the model size ratio and the collaboration frequency (more clearly evident for Qwen models than Pythia).
- Most of the LLMs/System 2 interventions are required at the start of the decoding and for tokens for which SLMs are uncertain.

**Strengths:**

- Proposes a new framework to analyze the collaborative behavior between models
- Empirical results shed new light on this collaborative behavior. In particular, the scaling law for collaboration and frequent positions of collaboration are quite interesting.

**Weaknesses:**

- The paper analyzes speculative decoding, contrastive decoding, and proxy tuning. Except for speculative decoding, it's not clear if the analysis provides any executable insights for the other two setups.
Drawing questionable analogies with human cognitive processes just because one model runs fast and the other slow and then commenting about how the collaborative distributions are different (L127-L129) is extremely flawed reasoning. The analogy doesn't make sense, except for the fact that one model is faster and the other is slower.

Comments about writing:
- Why is O_g being used and not O_f for p_f (fused logits) in Section 2.2
- L053: "allow" -> "allows"
- L195: "produce" -> "produced"

**Questions:**

- It is not clear what exactly is being illustrated in Figures 11, 12, and 13. What are the different features?
- How does one use the insights from this paper for contrastive decoding and proxy tuning?
- Currently, greedy decoding is used to establish whether collaboration is required or not. I wonder if the next token perplexity could be another measure.

---

> ### Author Response · Authors · 2024-11-22
> **Response - 1**
>
> We sincerely appreciate your positive feedback and the time and effort you have dedicated to reviewing our paper. Below, we provide further illustration and additional results to address your questions.
> ### Q1: Executable Insights for Contrastive Decoding and Proxy Tuning
> #### A1: Overview of Executable Framework
> Building on the insights from our findings, we propose a direct approach to optimizing the inference cost for both **Contrastive Decoding (CD)** and **Proxy Tuning (PT)**. Previous work on CD and PT typically involves collaboration across **all tokens** during text generation. However, our results suggest that this is unnecessary, as efficient collaboration can be achieved by focusing only on **specific tokens**.
>
> In this optimized framework:
> - **Small models serve as the main backbone** in CD and PT. They are tasked with generating the majority of the content during text generation.
> - **Token-level collaboration is determined dynamically** based on the logits distribution. Specifically, we identify whether a token requires collaboration from large models by analyzing the features of the match and mismatch logits between small models and mixed-scale models.
> To implement this, we can **train a lightweight token-level router** that leverages these logits features. The router determines when collaboration with a larger model is necessary, effectively balancing performance and efficiency.

---

> ### Author Response · Authors · 2024-11-22
> **Response - 2**
>
> #### A2: Preliminary Executable Results
> We present preliminary results for CD and PT, demonstrating their potential to optimize speed-performance trade-offs. Specifically, instead of conducting CD and PT on all tokens during text generation by small models, we focus these collaborations on a subset of mismatch tokens. By collaborating on these uncertain tokens alone, we achieve performance comparable to previous approaches that rely on collaborations for all tokens.
>
> - Table 1: Routing with Top-1 Token Logits of SLM for Contrastive Decoding
> 	- At each decoding step of the SLM, we determine whether to involve LLM collaboration based on the top-1 token logits of the SLM. A routing ratio of 0.0% implies decoding exclusively with the SLM, while a ratio of 100% indicates collaboration between the SLM and LLM at all steps.
> 	- Our results show that we can significantly reduce inference cost while maintaining comparable performance. Interestingly, when the performance gap between the SLM and LLM is small (e.g., 4B vs. 7B models), the performance improvement becomes less pronounced.
>
> | Contrastive Dec    | Threshold | 0.05  | 0.1   | 0.2   | 0.3   | 0.4   | 0.5    | 0.6    | 0.8    | 1.0  |
> | ------------------ | --------- | ----- | ----- | ----- | ----- | ----- | ------ | ------ | ------ | ---- |
> | Qwen1.5-0.5B w/ 7B | Ratio     | 0.0%  | 0.0%  | 0.3 % | 2.3 % | 5.6 % | 11.1 % | 17.2 % | 29.9 % | 100% |
> |                    | Accuracy  | 17.0  | 17.0  | 17.0  | 18.0  | 25.4  | 31.0   | 34.8   | 48.2   | 54.4 |
> | Qwen1.5-1.8B w/ 7B | Ratio     | 0.0   | 0.0   | 0.2 % | 1.4 % | 4.1 % | 8.4 %  | 13.9 % | 25.4 % | 100% |
> |                    | Accuracy  | 36.2% | 36.2% | 38.8  | 38.2  | 37.2  | 41.0   | 43.4   | 49.4   | 53.2 |
> | Qwen1.5-4B w/ 7B   | Ratio     | 0.0   | 0.0   | 0.2 % | 1.3 % | 3.8 % | 8 %    | 13.2 % | 24.6 % | 100% |
> |                    | Accuracy  | 52.2  | 52.2  | 52.6  | 51.8  | 53.2  | 51.0   | 51.4   | 51.0   | 51.0 |
> - Table 2: Routing with Top-1 Token Logits of SLM for Proxy Tuning
> 	- The results for Proxy Tuning exhibit a similar trend to Contrastive Decoding. Here, a routing ratio of 0.0% denotes generation exclusively using small tuned models, while a ratio of 100% indicates generation involving both small tuned models and large base models.
>
> | Proxy Tuning       | Threshold | 0.05 | 0.1  | 0.2   | 0.3   | 0.4   | 0.5    | 0.6    | 0.8    | 1.0  |
> | :----------------- | :-------- | ---- | ---- | ----- | ----- | ----- | ------ | ------ | ------ | ---- |
> | Qwen1.5-0.5B w/ 7B | Ratio     | 0.0% | 0.0% | 0.3 % | 1.9 % | 5.4 % | 10.3 % | 16.4 % | 28.9 % | 100% |
> |                    | Accuracy  | 9.6  | 9.8  | 11.0  | 11.8  | 17.8  | 20.2   | 25.6   | 38.4   | 57.0 |
> | Qwen1.5-1.8B w/ 7B | Ratio     | 0.0% | 0.0% | 0.1 % | 1 %   | 2.9 % | 6.4 %  | 11.3 % | 21.9 % | 100% |
> |                    | Accuracy  | 33.0 | 33.0 | 33.0  | 35.4  | 39.0  | 37.8   | 44.0   | 50.2   | 57.0 |
> | Qwen1.5-4B w/ 7B   | Ratio     | 0.0% | 0.0% | 0.1 % | 1 %   | 3.1 % | 6.5 %  | 11.5 % | 21.7 % | 100% |
> |                    | Accuracy  | 45.6 | 45.6 | 45.0  | 46.6  | 48.8  | 51.0   | 52.0   | 53.4   | 56.8 |
>
> Additionally, these findings align with recent advancements in test-time compute scaling applications. The token-level uncertainty analysis in our work can also be applied to entropy-based decoding methods like Entropix, where high-entropy tokens can be handled similarly to uncertain tokens in small language models.
>
> In our experiments, we used a simple routing mechanism based on the top-1 token logits threshold. However, this can be extended by training a more sophisticated router with richer features during decoding, which has the potential to further improve the efficiency and effectiveness of collaborative decoding.

---

> ### Author Response · Authors · 2024-11-22
> **Response - 3**
>
> ### Q2: Discussion of the System 1 & System 2 Analogy
> In this work, we draw inspiration from the analogy of System 1 and System 2, simplifying their collaboration into **Fast and Slow thinking**. System 1 efficiently handles approximately 95% of routine tasks, while System 2 is reserved for deliberately addressing the remaining 5% of complex work [1]. Together, they demonstrate the power of **high-efficiency collaboration**.
>
> We adopt this high-efficiency motivation to model the collaborative decoding methods between **fast and slow** (or **small and large**) models. Our experimental findings (Findings 1 and 2) show that small, fast models generate approximately 80% of tokens during the answering process, while large, slow models contribute the remaining 20%.
>
> Looking forward, we aim to expand these collaborative mechanisms to **reasoner and knowledger models**, such as OpenAI’s o1 and GPT-4, which not only embody the fast/slow model paradigm but also represent intuitive and deliberate thinking. Preliminary experiments reinforce our findings, indicating successful collaboration between o1 and existing large language models.
>
>
> [1] Booch, Grady, et al. "Thinking fast and slow in AI." Proceedings of the AAAI Conference on Artificial Intelligence. Vol. 35. No. 17. 2021.
>
>
> ### Q3: Response to Comments about Writing
> In Section 2.2, we use  $O_g$  to represent the **golden** outputs generated by mix-scaled language models. We agree that the suggestion to use  $O_f$  would provide better consistency with the **fused** logits. We will update this notation to enhance clarity and improve the reader’s understanding.
> Additionally, we appreciate you pointing out the typographical errors, and we will correct them in the updated manuscript.

---

> ### Author Response · Authors · 2024-11-22
> **Response - 4**
>
> ### Q4: Additional Illustration in Figures 11, 12, and 13
> In Figures 11, 12, and 13, we visualize the logits of the top 1 and top 5 tokens in the vocabulary of small models at each generation step. These logits are categorized into two distinct clusters:
> - **Matched tokens:** Tokens where the small model’s predictions align with those of the mix-scale model.
> - **Mismatched tokens:** Tokens where the small model’s predictions diverge from those of the mix-scale model.
>
> The visualization highlights that these clusters are separable, which supports our conclusion that **dynamic routing** can be implemented. Specifically, the uncertainty of token decoding in SLMs can guide the decision of whether to engage collaboration with larger models on a token-by-token basis.
> We utilize the following metrics to evaluate the correlation between matched and mismatched token logits:
>
> - **Silhouette Coefficient (SC)**
> 	- This metric (range: -1 to 1) assesses clustering quality by comparing intra-cluster cohesion and inter-cluster separation. Values > 0.5 indicate strong clustering performance.
> 	- A high SC value derived from Pearson or Spearman correlation demonstrates that the metric aligns well with the data.
> - **Davies-Bouldin Index (DBI)**
> 	- The DBI (range: $[0, ∞)$) measures clustering compactness and separation, where lower values (<1) suggest better clustering quality.
> 	- A low DBI derived from correlation methods indicates effective uncertainty estimation.
> - **Mean Cluster Center Distance (MCCD)**
> 	- MCCD measures the separation between cluster centers, with larger values indicating better distinction. Correlation methods that amplify these distances demonstrate their alignment with the data.
>
> - Table 3: Correlation Between Match/Mismatch Tokens and Top-K Token Logits of SLMs
> 	- Our results demonstrate the effectiveness of uncertainty estimation:
> 		- SC values are consistently close to 0.5.
> 		- DBI values are below 1, indicating compact and well-separated clusters.
> 		- MCCD values range between 10–20, reflecting robust inter-cluster distinction.
> 	- An exception is observed with Pythia series models, likely due to their insufficient pretraining.
>
> | Models  | Metric | GSM8k    |         | MMLU     |         | MBPP     |         |
> | ------- | ------ | -------- | ------- | -------- | ------- | -------- | ------- |
> |         |        | 5 tokens | 1 token | 5 tokens | 1 token | 5 tokens | 1 token |
> | Qwen1.5 | SC     | 0.465    | 0.503   | 0.445    | 0.457   | 0.47     | 0.469   |
> |         | DBI    | 0.806    | 0.805   | 0.838    | 0.917   | 0.772    | 0.909   |
> |         | MCCD   | 7.533    | 18.176  | 11.036   | 15.64   | 13.431   | 16.156  |
> | Pythia  | SC     | 0.465    | 0.358   | 0.485    | 0.286   | 0.464    | 0.315   |
> |         | DBI    | 0.79     | 1.18    | 0.755    | 1.416   | 0.779    | 1.3     |
> |         | MCCD   | 21.584   | 14.125  | 22.584   | 16.289  | 21.325   | 16.843  |

---

> ### Author Response · Authors · 2024-11-22
> **Response - 5**
>
> ### Q5: Greedy Decoding and Next-Token Perplexity for Collaboration Measuring
> #### A1: Overview of Different Metrics
> This is an excellent question, and we appreciate the opportunity to address it. Below, we analyze the relationship between different metrics and provide comparative results.
> At each decoding step, we obtain logits over the vocabulary from the SLMs. These logits are normalized into the range $[0, 1]$ using the `softmax` function. This gives the next-token probabilities:
> $$P(y_t) = \text{softmax}(\text{logits}_t)$$
> During greedy decoding, the next token $y_t$ is selected as the one with the highest probability:
> $$y_t = \arg \max_i^V P(y_t^i)$$
> where $V$  is the size of the vocabulary.
>
>
> Below we will provide an analysis of the next token entropy and perplexity:
> - **Next token entropy**, $H(y_t)$, is defined as $H(y_t) = -\sum_{i=1}^V P(y_t^i) \log P(y_t^i)$, which quantifies the uncertainty of the model’s prediction at step t. Lower entropy indicates more confident predictions, while higher entropy suggests greater uncertainty in selecting the next token.
> - **Perplexity**, when based on cross-entropy, requires access to the golden tokens (ground truth sequence). Since our routing analysis does not have access to golden tokens, we instead rely directly on entropy as a proxy for measuring uncertainty.
> 	- **Next token perplexity** is defined as $\text{Perplexity}(y_t) = 2^{H(y_t)}$, which transforms entropy into an interpretable measure representing the average branching factor of the model’s distribution over the next token. A lower perplexity implies a narrower, more confident distribution.
> 	- **Sequence perplexity**, on the other hand, measures the uncertainty over the entire sequence and is defined as $\text{Perplexity(sequence)} = 2^{-\frac{1}{T} \sum_{t=1}^T \log P(y_t)}$, where T is the length of the sequence. Sequence perplexity can be seen as the geometric mean of next token perplexities across all decoding steps.
>
> The relationship among these metrics is intrinsic: next token logits determine next token entropy and perplexity through their normalized probabilities. Higher entropy and perplexity indicate a more uniform distribution, while lower values suggest peaked distributions. Sequence perplexity aggregates these effects over all steps, offering a global view of the model’s predictive confidence across the sequence.

---

> > ### Author Response · Authors · 2024-11-22
> > **Response - 6**
> >
> > #### A2: Additional Results on Different Metrics
> > To further evaluate the effectiveness of various metrics for routing from SLMs to mix-scaled models, we conducted additional analysis. Building on the executable insights provided in Q1, which demonstrated the effectiveness of routing and quantitative uncertainty scores using clustering metrics, we extended our investigation to entropy and perplexity metrics.
> >
> > We analyzed the correlation between token matching/mismatching and entropy/perplexity scores, expanding on Findings 4 presented in Figures 11, 12, and 13.
> >
> > - Table 4: Correlation Between Match/Mismatch Tokens and Entropy/Perplexity Scores of SLMs (Qwen series)
> > 	- The results presented in Table 4 reveal trends similar to those observed in the previous analysis in **Q3**. Furthermore, entropy and perplexity metrics perform better on recognizing mismatched tokens.
> > 	- This consistency underscores the effectiveness of entropy and perplexity metrics, demonstrating that they serve a similar role to the top- k  token logits in identifying match and mismatch tokens.
> > 	- These findings align closely with our analysis in **A1**, further validating the utility of entropy and perplexity as reliable metrics for guiding collaborative decoding decisions.
> > |Task|GPQA|||IFEval|||MedQA|||
> > |---|---|---|---|---|---|---|---|---|---|
> > |Feature/Metric|SC|DBI|MCCD|SC|DBI|MCCD|SC|DBI|MCCD|
> > |Top logits of 1 token|0.572|0.668|13.182|0.317|1.22|117.359|0.308|1.261|123.797|
> > |Top logits of 5 tokens|0.45|0.896|5.86|0.26|1.473|107.906|0.216|1.677|105.803|
> > |Token Entropy|0.742|0.456|2.667|0.624|0.536|3.662|0.632|0.563|2.445|
> > |Token PPL|0.838|0.504|3.261|0.767|0.53|5.768|0.775|0.553|2.986|
> > |Context PPL|0.934|0.325|2.276|0.588|0.61|0.603|0.569|0.597|0.264|
> >
> > Our results confirm the feasibility of implementing routing using entropy and perplexity scores, aligning with recent developments in entropy-based decoding projects, such as **Entropix** [1]. We believe our findings provide valuable insights and can further advance research in this area.
> >
> > [1] https://github.com/xjdr-alt/entropix
> >
> > -----
> >
> > Thank you again for your feedback and queries. We welcome any further discussion to address potential misunderstandings or to clarify our results.

---

> > > ### Comment · Reviewer_gboG · 2024-11-24
> > >
> > > Thanks for sharing these insights. While I appreciate sharing these additional insights during rebuttal, I agree with reviewer VTMo that the need for so many new experiments means that the original paper lacked details and some obvious ablations. Even with the rebuttal, in the Response-2 I have no idea what the threshold is and what is the accuracy? While these can be clarified again in the rebuttal phase, this constant back-and-forth is a sign that the authors are not careful while sharing the results.
> > >
> > > Finally, on a technical note, I realized a big mistake that I had missed earlier in my reading. Logits refer to the unnormalized scores. So the use of logit throughout the paper is technically wrong! Even in the contrastive decoding paper, they use log-probability instead of logit.
> > >
> > > **Overall, I'm leaning negative now and have reduced score by a point. The paper needs substantial revision, and it would be better if the authors just resubmitted it to another venue because there are too many holes in the current manuscript.** I would also suggest reducing the emphasis on cognitive science analogies, especially when the two comparables bear little resemblance.

---

> > > > ### Author Response · Authors · 2024-11-25
> > > >
> > > > Thank you for your thoughtful comments and for taking the time to review our work in detail. We would like to address potential misunderstandings in our previous response and clarify key points from the paper.
> > > >
> > > > Our motivation for **including additional experiments was to provide broader insights and explore the broader impact of our findings**. We apologize if this has added to the burden of the rebuttal process. However, we want to emphasize that these experiments are not intended to suggest a lack of detail or obvious ablations in our core contributions.
> > > >
> > > > The central focus of our paper lies in the findings derived from **three methods applied to three datasets across two model series**, which we believe are sufficiently self-contained. **The additional experiments discussed in the rebuttal serve to extend and contextualize these findings, offering new avenues for future work.** To clarify:
> > > > - **Q1** provides executable insights and new results on contrastive decoding and proxy tuning. Similarly, **Q5** includes additional experiments on router optimization, extending **Finding 4** and building on the discussion in Section 5 ("Cost-Aware Collaboration Optimization").
> > > > -  **Q4** relates to the source data for visualizations in Figures 11 and 12, which were excluded from the main paper due to space limitations. These do not represent new experiments but rather provide supplementary information.
> > > > - **Q2** offers detailed responses and further elaboration on the motivation discussed in the Introduction (Lines 84–98).
> > > >
> > > > ### Use of the Term "Logit"
> > > > We acknowledge that the term “logit” may have caused some confusion. Our intent was to compare the top-1 token selected by the SLM and the SLM+LLM models under greedy decoding. While the final token is obtained using $\text{argmax}(\text{logits})$, this is effectively equivalent to using probabilities after softmax.
> > > > For contrastive decoding, we primarily refer to the implementation in [1]. We note that Section 2.2 contains an incorrect citation for this reference, which we will correct. This approach uses unnormalized scores (logits) directly, as assigned by the amateur and expert models.
> > > > ### Description of Thresholds and Accuracy
> > > > Our exploration of token uncertainty, as mentioned in Section 4.2.2, aligns with the discussion in Section 4 ("Cost-Aware Collaboration Optimization"). Here, we cite [2] to recommend dynamic collaboration (also definition of threshold) based on heuristic rules. The additional experiments in this area were conducted to further explore the broader implications of these findings.
> > > > ### Reducing Emphasis on Cognitive Science Analogies
> > > > While we used the "System 1 and System 2" framework to illustrate fast and slow behaviors, our primary focus was on their high-efficiency collaboration, which is central to our work. This approach aligns with prior research, such as [3], but we will revise these descriptions in the paper to ensure clarity and focus.
> > > >
> > > > ------
> > > >
> > > > Once again, we sincerely thank you for your detailed and constructive feedback. While we will revise our paper to address the noted areas of potential confusion, we respectfully maintain that there are no substantial changes to our main experiments or findings.
> > > >
> > > > We welcome further discussion on these concerns and are committed to refining our work in response to your valuable suggestions. Thank you again for your thoughtful review and insights.
> > > >
> > > > [1] O'Brien, Sean, and Mike Lewis. "Contrastive decoding improves reasoning in large language models." _arXiv preprint arXiv:2309.09117_ (2023).
> > > >
> > > > [2] Kim, Sehoon, et al. "Speculative decoding with big little decoder." _Advances in Neural Information Processing Systems_ 36 (2024).
> > > >
> > > > [3] Lin, Bill Yuchen, et al. "Swiftsage: A generative agent with fast and slow thinking for complex interactive tasks." _Advances in Neural Information Processing Systems_ 36 (2024).

---

### Official Review · Reviewer_zvri · 2024-11-06

**Soundness:** 2
**Presentation:** 3
**Contribution:** 3
**Rating:** 6
**Confidence:** 3

**Summary:**

The paper studies collaborative decoding, where small language models and large language models work together in the decoding process. In particular, the paper offers a unifying perspective on 3 different collaborative decoding techniques: proxy tuning, speculative decoding and contrastive decoding. Authors categorize the larger model as System 2 and smaller model as system 1.
The paper studies the 3 techniques, their commonalities and differences through their framework FS-GEN (Fast and Slow Generating).
They find that only small fraction of decoding steps require collaboration and that System 1 and 2 follow a scaling law related to parameter ratios.

**Strengths:**

Paper studies a relatively under explored but important and emerging area of research.
The findings are interesting, particularly the 2:8 law, collaborations being most necessary at the beginning of decoding and that high uncertainty tokens within System 1 are more likely to require collaboration.
Some of the findings could spur targeted research in the field of collaborative decoding.
Experimental benchmarks cover different capabilities like knowledge, math and coding, as well as two LLM families.

**Weaknesses:**

The System 1 and System 2 analogy is not well fleshed out, to the point where it feels more like a distraction from the main contributions.

The line fits on the param ratio scaling plot aren't very convincing.

The uncertainty analysis is only qualitative - quantitative metrics to support this hypothesis (covering different tasks and model families) are missing. Without them its hard to have confidence in this finding.

**Questions:**

Related work is pushed to the Appendix. This is a strange choice. I understand there might have been a space crunch, but Related Work makes much more sense to be in the main paper.

---

> ### Author Response · Authors · 2024-11-22
> **Response - 1**
>
> We sincerely appreciate your positive feedback, along with the time and effort you have put into reviewing our paper.
> We would like to give our thoughts and new results to address your concerns.
>
> ### Q1: Relationship Between Main Contributions and the System 1 & System 2 Analogy
> In this work, we draw inspiration from the analogy of System 1 and System 2, simplifying their collaboration into **Fast and Slow thinking**. System 1 efficiently handles approximately 95% of routine tasks, while System 2 is reserved for deliberately addressing the remaining 5% of complex work [1]. Together, they demonstrate the power of **high-efficiency collaboration**.
>
> We adopt this high-efficiency motivation to model the collaborative decoding methods between **fast and slow** (or **small and large**) models. Our experimental findings (Findings 1 and 2) show that small, fast models generate approximately 80% of tokens during the answering process, while large, slow models contribute the remaining 20%.
>
> Looking forward, we aim to expand these collaborative mechanisms to **reasoner and knowledger models**, such as OpenAI’s o1 and GPT-4, which not only embody the fast/slow model paradigm but also represent intuitive and deliberate thinking. Preliminary experiments reinforce our findings, indicating successful collaboration between o1 and existing large language models.
>
>
> [1] Booch, Grady, et al. "Thinking fast and slow in AI." Proceedings of the AAAI Conference on Artificial Intelligence. Vol. 35. No. 17. 2021.
>
>
> ### Q2: Line Fitting on the Parameter Ratio Scaling
> The line fitting in Figures 4 and 5 is influenced by both **data size** and **model performance**:
> - **Data Size:** Due to computational limitations, we sampled only ~500 data points for each task. This sampling constraint may contribute to fluctuations in the observed curve.
> - **Model Performance:** Parameter ratio scaling laws are significantly affected by model performance. While Qwen series models maintain consistent performance, Pythia models underperform due to insufficient pretraining, thereby affecting the collaboration dynamics between large and small models.
>
> To further validate our findings, we conducted additional experiments on **OpenELM models** [2], which exhibit better performance compared to Pythia.
>
> - Table 1: Line Fitting Results of OpenELM Models
> 	- Given the following formula of scale ratio law $$CoF_{\text{lower}}=\gamma \cdot {R}^{-\alpha}+\beta$$where $R=\frac{N_l}{N_s}$, we compute the coefficients with model parameters and collaboration frequency.
> 	- This table presents the line fitting results for oracle decoding, including fitting error, fitting coefficients, and the final x (i.e., $R^{-\alpha}$)  and  y  values on the fitting curve.
> 	- The results demonstrate a strong fitting effect, confirming the generalizability of our findings across different model families.
> 	- These results further indicate that the performance of collaborative decoding is influenced by the underlying model’s performance.
>
> | Task      | Formula & Fitting Error                     | Coordinates | 270M/450M | 450M/1.1B | 1.1B/3B | 270M/1.1B | 450M/3B | 270M/3B |
> | --------- | ------------------------------------------- | ----------- | --------- | --------- | ------- | --------- | ------- | ------- |
> |           |                                             | Ratio       | ≈1.67     | ≈2.44     | ≈2.73   | ≈4.07     | ≈6.67   | ≈11.11  |
> | GSM8k     | $\gamma=4.66, \alpha=-0.0041, \beta=4.67$   | X axis      | 0.9979    | 0.9964    | 0.9959  | 0.9943    | 0.9923  | 0.9902  |
> |           | MSE Loss = 1.16e-6                          | Y axis      | 0.0250    | 0.0320    | 0.0320  | 0.0420    | 0.0490  | 0.0610  |
> | MMLU-STEM | $\gamma=5.02, \alpha=-0.0022, \beta=5.04$   | X axis      | 0.9989    | 0.9981    | 0.9978  | 0.9969    | 0.9959  | 0.9948  |
> |           | MSE Loss = 2.25e-6                          | Y axis      | 0.0280    | 0.0300    | 0.0350  | 0.0350    | 0.0420  | 0.0490  |
> | MBPP      | $\gamma=23.83, \alpha=-0.0007, \beta=23.84$ | X axis      | 0.9996    | 0.9994    | 0.9993  | 0.9990    | 0.9987  | 0.9983  |
> |           | MSE Loss = 7.04e-5                          | Y axis      | 0.0220    | 0.0170    | 0.0380  | 0.0200    | 0.0480  | 0.0500  |
>
> [2] Mehta, Sachin, et al. "Openelm: An efficient language model family with open-source training and inference framework." _arXiv e-prints_ (2024): arXiv-2404.

---

> ### Author Response · Authors · 2024-11-22
> **Response - 2**
>
> ### Q3: Quantitative Metrics for Uncertainty Analysis
> To strengthen the evidence supporting our uncertainty analysis, we provide additional quantitative results, generalizing across all model combinations and methods. We utilize the following metrics to evaluate the correlation between matched and mismatched token logits:
>
> - **Silhouette Coefficient (SC)**
> 	- This metric (range: -1 to 1) assesses clustering quality by comparing intra-cluster cohesion and inter-cluster separation. Values > 0.5 indicate strong clustering performance.
> 	- A high SC value derived from Pearson or Spearman correlation demonstrates that the metric aligns well with the data.
> - **Davies-Bouldin Index (DBI)**
> 	- The DBI (range: $[0, ∞)$) measures clustering compactness and separation, where lower values (<1) suggest better clustering quality.
> 	- A low DBI derived from correlation methods indicates effective uncertainty estimation.
> - **Mean Cluster Center Distance (MCCD)**
> 	- MCCD measures the separation between cluster centers, with larger values indicating better distinction. Correlation methods that amplify these distances demonstrate their alignment with the data.
>
> - Table 2: Correlation Between Match/Mismatch Tokens and Top-K Token Logits of SLMs
> 	- Our results demonstrate the effectiveness of uncertainty estimation:
> 		- SC values are consistently close to 0.5.
> 		- DBI values are below 1, indicating compact and well-separated clusters.
> 		- MCCD values range between 10–20, reflecting robust inter-cluster distinction.
> 	- An exception is observed with Pythia series models, likely due to their insufficient pretraining.
>
> | Models  | Metric | GSM8k    |         | MMLU     |         | MBPP     |         |
> | ------- | ------ | -------- | ------- | -------- | ------- | -------- | ------- |
> |         |        | 5 tokens | 1 token | 5 tokens | 1 token | 5 tokens | 1 token |
> | Qwen1.5 | SC     | 0.465    | 0.503   | 0.445    | 0.457   | 0.47     | 0.469   |
> |         | DBI    | 0.806    | 0.805   | 0.838    | 0.917   | 0.772    | 0.909   |
> |         | MCCD   | 7.533    | 18.176  | 11.036   | 15.64   | 13.431   | 16.156  |
> | Pythia  | SC     | 0.465    | 0.358   | 0.485    | 0.286   | 0.464    | 0.315   |
> |         | DBI    | 0.79     | 1.18    | 0.755    | 1.416   | 0.779    | 1.3     |
> |         | MCCD   | 21.584   | 14.125  | 22.584   | 16.289  | 21.325   | 16.843  |
>
> ### Q4: Position of Related Works in the Paper
> We appreciate your feedback regarding the placement of related works. To improve clarity, we will reorganize the structure of the paper. Specifically:
> - **Revised structure:** We will simplify the presentation of the main text to enhance readability.
> - **Related works:** We will introduce related work briefly in the main text, ensuring it is more integrated and accessible.
>
> We hope these responses address your concerns and provide further clarity. Thank you once again for your constructive feedback and valuable suggestions, which have been instrumental in improving our work.

---

### Official Review · Reviewer_DHMi · 2024-11-06

**Soundness:** 3
**Presentation:** 3
**Contribution:** 4
**Rating:** 6
**Confidence:** 4

**Summary:**

The paper explores collaborative decoding strategies between large language models (LLMs) and small language models (SLMs). The authors introduce the FS-GEN framework, categorizing LLMs as System 2 (slow and deliberate) and SLMs as System 1 (fast and intuitive). The research focuses on decoding methods like speculative decoding, contrastive decoding, and proxy tuning to improve efficiency and mitigate issues like high inference time.

**Strengths:**

Originality:- The paper introduces a novel FS-GEN framework.
Quality:- The tables and figures are very well used.
The paper is written with a great clarity.
significance:- The paper compares from smaller models to larger ones, based on the number of parameters.

**Weaknesses:**

Could provide more discussion of practical applications.
Trade-offs between the inference time and cost can be a great addition.
The experiments focused on only few tasks like:- MMLU-STEM, GSM8k, and MBPP, Having experiments over domain specific datasets can give a better understanding.

**Questions:**

How generalizable do you believe your findings are to other language tasks or domains?
How do you think the collaborative patterns might change, If different sampling technique is used.

---

> ### Author Response · Authors · 2024-11-22
> **Response - 1**
>
> We sincerely thank you for your positive feedback and valuable suggestions. Below, we provide our thoughts and new results addressing your questions.
> ### Q1: Discussion of Practical Applications
> The primary motivation behind collaborative decoding between large and small language models is to optimize the speed-performance trade-off. Previous works, such as speculative decoding, have demonstrated the effectiveness of reducing inference time. Our work generalizes this collaboration to broader methods, including **Contrastive Decoding (CD)** and **Proxy Tuning (PT)**.
>
> Here, we present some preliminary results for CD and PT, demonstrating their potential to optimize speed-performance trade-offs. Specifically, instead of conducting CD and PT on all tokens during text generation by small models, we focus these collaborations on a subset of mismatch tokens compared to mix-scaled models. By collaborating on these uncertain tokens alone, we achieve performance comparable to previous approaches that rely on collaborations for all tokens.
>
> - Table 1: Routing with Top-1 Token Logits of SLM for Contrastive Decoding
> 	- At each decoding step of the SLM, we determine whether to involve LLM collaboration based on the top-1 token logits of the SLM. A routing ratio of 0.0% implies decoding exclusively with the SLM, while a ratio of 100% indicates collaboration between the SLM and LLM at all steps.
> 	- Our results show that we can significantly reduce inference cost while maintaining comparable performance. Interestingly, when the performance gap between the SLM and LLM is small (e.g., 4B vs. 7B models), the performance improvement becomes less pronounced.
>
> | Contrastive Decoding | Threshold | 0.05  | 0.1   | 0.2   | 0.3   | 0.4   | 0.5    | 0.6    | 0.8    | 1.0  |
> | -------------------- | --------- | ----- | ----- | ----- | ----- | ----- | ------ | ------ | ------ | ---- |
> | Qwen1.5-0.5B w/ 7B   | Ratio     | 0.0%  | 0.0%  | 0.3 % | 2.3 % | 5.6 % | 11.1 % | 17.2 % | 29.9 % | 100% |
> |                      | Accuracy  | 17.0  | 17.0  | 17.0  | 18.0  | 25.4  | 31.0   | 34.8   | 48.2   | 54.4 |
> | Qwen1.5-1.8B w/ 7B   | Ratio     | 0.0   | 0.0   | 0.2 % | 1.4 % | 4.1 % | 8.4 %  | 13.9 % | 25.4 % | 100% |
> |                      | Accuracy  | 36.2% | 36.2% | 38.8  | 38.2  | 37.2  | 41.0   | 43.4   | 49.4   | 53.2 |
> | Qwen1.5-4B w/ 7B     | Ratio     | 0.0   | 0.0   | 0.2 % | 1.3 % | 3.8 % | 8 %    | 13.2 % | 24.6 % | 100% |
> |                      | Accuracy  | 52.2  | 52.2  | 52.6  | 51.8  | 53.2  | 51.0   | 51.4   | 51.0   | 51.0 |
> - Table 2: Routing with Top-1 Token Logits of SLM for Proxy Tuning
> 	- The results for Proxy Tuning exhibit a similar trend to Contrastive Decoding. Here, a routing ratio of 0.0% denotes generation exclusively using small tuned models, while a ratio of 100% indicates generation involving both small tuned models and large base models.
>
> | Proxy Tuning       | Threshold | 0.05 | 0.1  | 0.2   | 0.3   | 0.4   | 0.5    | 0.6    | 0.8    | 1.0  |
> | :----------------- | :-------- | ---- | ---- | ----- | ----- | ----- | ------ | ------ | ------ | ---- |
> | Qwen1.5-0.5B w/ 7B | Ratio     | 0.0% | 0.0% | 0.3 % | 1.9 % | 5.4 % | 10.3 % | 16.4 % | 28.9 % | 100% |
> |                    | Accuracy  | 9.6  | 9.8  | 11.0  | 11.8  | 17.8  | 20.2   | 25.6   | 38.4   | 57.0 |
> | Qwen1.5-1.8B w/ 7B | Ratio     | 0.0% | 0.0% | 0.1 % | 1 %   | 2.9 % | 6.4 %  | 11.3 % | 21.9 % | 100% |
> |                    | Accuracy  | 33.0 | 33.0 | 33.0  | 35.4  | 39.0  | 37.8   | 44.0   | 50.2   | 57.0 |
> | Qwen1.5-4B w/ 7B   | Ratio     | 0.0% | 0.0% | 0.1 % | 1 %   | 3.1 % | 6.5 %  | 11.5 % | 21.7 % | 100% |
> |                    | Accuracy  | 45.6 | 45.6 | 45.0  | 46.6  | 48.8  | 51.0   | 52.0   | 53.4   | 56.8 |
>
> Additionally, these findings align with recent advancements in test-time compute scaling applications. The token-level uncertainty analysis in our work can also be applied to entropy-based decoding methods like Entropix [1], where high-entropy tokens can be handled similarly to uncertain tokens in small language models.
> In the above experiments, we used a simple routing mechanism based on the top-1 token logits threshold. However, this can be extended by training a more sophisticated router with richer features during decoding, which has the potential to further improve the efficiency and effectiveness of collaborative decoding.
>
> [1] https://github.com/xjdr-alt/entropix

---

> ### Author Response · Authors · 2024-11-22
> **Response - 2**
>
> ### Q2: Experiments on Domain-Specific Datasets
> To validate the robustness of our findings, we conducted additional experiments on **GPQA**, **MedQA**, and **IFEval**, which encompass biology, medical, and physics question-answering tasks, as well as instruction-following tasks in open-domain.
>
> - Table 3: Results of  $CoF_{\text{lower}}$  on Additional Domain Tasks
> 	- The results indicate that  $CoF_{\text{lower}}$  is consistently below 20% across various methods, tasks, and model combinations. Furthermore, we observe a decreasing trend in  $CoF_{\text{lower}}$  as the ratio of model parameters decreases.
> 	- We also found that the collaboration rate of general models on domain tasks is slightly higher than that on general tasks.
>
> | Task               | GPQA  |       |       | IFEval |       |       | MedQA |       |       |
> | ------------------ | ----- | ----- | ----- | ------ | ----- | ----- | ----- | ----- | ----- |
> | Method / CoF_lower | SD    | CD    | PT    | SD     | CD    | PT    | SD    | CD    | PT    |
> | Qwen1.5-0.5B w/ 7B | 0.162 | 0.211 | 0.157 | 0.208  | 0.298 | 0.2   | 0.23  | 0.296 | 0.225 |
> | Qwen1.5-1.8B w/ 7B | 0.13  | 0.198 | 0.133 | 0.174  | 0.238 | 0.164 | 0.194 | 0.314 | 0.19  |
> | Qwen1.5-4B w/ 7B   | 0.099 | 0.155 | 0.098 | 0.149  | 0.221 | 0.145 | 0.169 | 0.308 | 0.165 |
>
> When extending model collaborations from generalist to specialist tasks, we anticipate that the collaboration frequency will decrease due to the narrower distribution of domain-specific terminology. However, the lack of a comprehensive range of specialized model series limits further analysis at this stage, and we leave this exploration as future work.
>
> ### Q3: Different Sampling Techniques
> In our current work, we use **greedy decoding** to compute the matching rate of tokens between small and large language models. This choice aligns with our initial motivation of achieving collaborative decoding with minimal intervention in small models, treating the collaborative decoding results as golden tokens.
>
> For scenarios where exact matching is less critical and the focus shifts to performance-speed optimization, other sampling techniques can be explored. These techniques might yield better performance with reduced collaboration frequency, leading to more efficient collaborations. However, quantifying results becomes more challenging due to the increased uncertainty introduced by sampling.
> We believe this is an exciting direction for future research, as it opens up possibilities for balancing efficiency and performance through alternative decoding strategies.

---

> > ### Comment · Reviewer_DHMi · 2024-11-28
> >
> > The questions are clearly explained and clarified, for the 2nd question, I wanted to know if you have tried for other domains which were not included in the paper. But overall I am satisfied with the work.

---

> > > ### Author Response · Authors · 2024-11-29
> > >
> > > Thank you for your positive and encouraging feedback. In the original paper, we explored domains such as mathematics (GSM8k), code (MBPP), and general knowledge (MMLU). In the rebuttal, we expanded our analysis to include additional domains, such as medical knowledge (MedQA) and physical/chemical/biological sciences (GPQA). The results from these new domains consistently support our original findings.
> > >
> > > In conclusion, our study covers a broad range of common domains, and we are enthusiastic about extending our approach to explore other relevant domains in future work. Thank you once again for your valuable feedback and for the opportunity to further refine our work.

---

### Comment · Area_Chair_ypw2 · 2024-11-21
**Please respond and update the score if necessary**

Dear Reviewers,

Kindly ensure that you respond proactively to the authors' replies so we can foster a productive discussion. If necessary, please update your score accordingly. We greatly appreciate the time and effort you’ve dedicated to the review process, and your contributions are key to making this process run smoothly.

Thank you,

AC

---

### Author Response · Authors · 2024-11-26

Dear Reviewers,

Thank you for your comments on our paper. We have carefully revised the manuscript to address your concerns, incorporating additional explanations, analyses, and clarifications as needed. For your convenience, all new content in the revised manuscript is highlighted in blue. Below, we summarize the changes made in response to your comments:

### Descriptive Questions
- **Update to the analogy of System 1 and System 2 (@Reviewers zvri, gboG, VTMo):** We have reduced references to human cognition and instead emphasized the efficient collaboration between fast and slow systems. These updates are reflected in Figure 1 and the Introduction (Section 1).
- **Discussion on different sampling techniques (@Reviewers DHMi, gboG):** A new discussion of additional sampling techniques is provided in Appendix D.
- **Explanation of features in Figures 11, 12, and 13 (@Reviewer gboG):** We have added further analysis of these features in Section 6 (Discussion).
- **Simplified related works section (@Reviewers zvri, VTMo):** A concise version of the related works previously in the appendix has been rewritten and moved to Section 2 (Related Works).
- **Discussion of differences in datasets and models (@Reviewer VTMo):** We now discuss the impact of dataset size and model performance in Section 6 (Discussion).
- **Additional running example for experimental settings and reproducibility (@Reviewer VTMo):** A detailed example, focusing on computing collaboration frequency, is included in Appendix C (Table 2). The implementation code is provided in an anonymous repository.
- **Explanation of outputs and logits (@Reviewer gboG):** We have corrected typos, resolved citation errors, and provided further explanations regarding logits in Appendix D.

### Experimental Questions
- **Discussion on domain-specific datasets (@Reviewer DHMi):** We have added results on collaboration frequency for MedQA, GPQA, and IFEval datasets in Appendix E1 (Table 3).
- **Discussion on practical and executable applications (@Reviewers DHMi, gboG):** We provided additional results on token-based routing using SLM logits to improve quality-efficiency trade-offs. These updates are included in Section 6 (Discussion, Figure 11) and Appendix F.2 (Figures 17, 18).
- **Further analysis of parameter ratio scaling effects (@Reviewer zvri):** We analyzed cases of poor fitting and updated more results for OpenELMs in Section 5.1.2 (Figure 5) and Appendix E1.
- **Quantitative metrics for uncertainty analysis (@Reviewer zvri):** We included corresponding quantitative metrics for Figures 10 and 11 in Table 4 and provided a detailed explanation of the correlations in Appendix F.1.

----
Our core findings remain unchanged, but we have clarified key points, validated our results on domain-specific datasets, and supplemented our discussion on the practical application of our empirical results. We hope these updates sufficiently address your concerns. Thank you for your time and continued consideration.

---

### Meta-Review · Area_Chair_ypw2 · 2024-12-20

**Metareview:**

The paper examines collaborative decoding, where small and large language models work together during the decoding process. It unifies three techniques: proxy tuning, speculative decoding, and contrastive decoding, framing them through the FS-GEN (Fast and Slow Generating) framework. The larger model is characterized as System 2, which operates slowly and deliberately, while the smaller model is System 1, functioning quickly and intuitively. The study finds that only a small fraction of decoding steps require collaboration and identifies a scaling law related to parameter ratios. Using the Qwen and Pythia series, evaluated across datasets like MMLU-STEM, GSM8k, and MBPP, the research highlights that collaborative interactions are most critical at the start of the generation process, with an optimal interaction frequency around an 80-20 ratio, depending on the task.

My decision is to reject the paper, as it requires substantial revisions, particularly in areas highlighted by Reviewer VTMo. Additionally, the authors need to clarify the terminology used throughout the paper.

**Additional Comments On Reviewer Discussion:**

Reviewer gboG identified a significant oversight in the paper regarding the use of "logits," which should correctly refer to unnormalized scores. The proper term, as used in the contrastive decoding paper, is "log-probability," leading to gboG lowering their score. Reviewer DHMi also reduced their score from 8 to 6, although this score is notably an outlier compared to other review scores. Reviewer VTMo stated that the paper needs substantial revisions to be both understandable and assessable in terms of its merits. The paper lacks detailed experimental information, and its conclusions are inadequately explained. Additionally, the authors have not addressed these issues.

---

### Decision · Program_Chairs · 2025-01-22

Reject